# INFORMATION-THEORETIC ONLINE MEMORY SELECTION FOR CONTINUAL LEARNING

**Shengyang Sun**[*1], **Daniele Calandriello**[4], **Huiyi Hu**[*2], **Ang Li**[*3], **Michalis K. Titsias**[4]

[1]University of Toronto, [1]Vector Institute, [2]Google Brain, [3]Baidu Apollo, [4]DeepMind
ssy@cs.toronto.edu, angli01@baidu.com,
{dcalandriello, clarahu, mtitsias}@google.com

## ABSTRACT

A challenging problem in task-free continual learning is the online selection of a representative replay memory from data streams. In this work, we investigate the online memory selection problem from an information-theoretic perspective. To gather the most information, we propose the *surprise* and the *learnability* criteria to pick informative points and to avoid outliers. We present a Bayesian model to compute the criteria efficiently by exploiting rank-one matrix structures. We demonstrate that these criteria encourage selecting informative points in a greedy algorithm for online memory selection. Furthermore, by identifying the importance of *the timing to update the memory*, we introduce a stochastic information-theoretic reservoir sampler (InfoRS), which conducts sampling among selective points with high information. Compared to reservoir sampling, InfoRS demonstrates improved robustness against data imbalance. Finally, empirical performances over continual learning benchmarks manifest its efficiency and efficacy.

## 1 INTRODUCTION

Continual learning (Robins, 1995; Goodfellow et al., 2013; Kirkpatrick et al., 2017) aims at training models through a non-stationary data stream without catastrophic forgetting of past experiences. Specifically, replay-based methods (Lopez-Paz & Ranzato, 2017; Rebuffi et al., 2017; Rolnick et al., 2019) tackle the continual learning problem by keeping a replay memory for rehearsals over the past data. Given the limited memory budget, selecting a representative memory becomes critical. The majority of existing approaches focus on task-based continual learning and update the memory based on the given task boundaries. Since the requirement for task boundaries is usually not realistic, general continual learning (GCL) (Aljundi et al., 2019a; Delange et al., 2021; Buzzega et al., 2020) has received increasing attention, which assumes that the agent observes the streaming data in an online fashion without knowing task boundaries. GCL makes the online memory selection more challenging since one needs to update the memory in each iteration based only on instant observations. So, successful memory management for GCL needs to be both efficient and effective.

Due to the online constraint, only a few memory selection methods are applicable to GCL. A selection method compatible with GCL is reservoir sampling (RS) (Vitter, 1985). In particular, RS can be used to randomly extract a uniform subset from a data stream in a single pass, and it showed competitive performances for GCL for balanced streams (Chaudhry et al., 2019b; Buzzega et al., 2020). However, when the data stream is imbalanced, RS tends to miss underrepresented (but essential) modes of the data distribution. To improve the robustness against data imbalance, gradient-based sample selection (GSS) (Aljundi et al., 2019b) proposes to minimize the gradient similarities in the memory, but evaluating many memory gradients in each iteration incurs high computational costs.

We investigate online memory selection from an information-theoretic perspective, where the memory attempts to maintain the most information for the underlying problem. Precisely, we evaluate each data point based on the proposed *surprise* and *learnability* criteria. Surprise captures how unexpected a new point is given the memory, and allows us to include new information in the memory. In contrast, learnability captures how much of this new information can be absorbed without inter-

---

[*]Work done in DeepMind.

ference, allowing us to avoid outliers. Then we present a scalable Bayesian model that can compute surprise and learnability with a small computational footprint by exploiting rank-one matrix structures. Finally, we demonstrate the effectiveness of the proposed criteria using a greedy algorithm.

While keeping a representative memory is essential, we show that the *timing* of the memory updates can also be crucial for continual learning performance. Concretely, we highlight that the agent should not update the memory as soon as it sees new data, otherwise it might prematurely remove historical data from the memory and weaken the replay regularization in the GCL process. This phenomenon affects the greedy algorithm much more than RS, since the memory updates in RS appear randomly over the whole data stream. To combine the merits of information-theoretic criteria and RS, we modify reservoir sampling to select informative points only. This filters out uninformative points, and thus encourages a diverse memory and improves the robustness against data imbalance.

Empirically, we demonstrate that the proposed information-theoretic criteria encourage to select representative memories for learning the underlying function. We also conduct standard continual learning benchmarks and demonstrate the advantage of our proposed reservoir sampler over strong GCL baselines at various levels of data imbalance. Finally, we illustrate the efficiency of the proposed algorithms by showing their small computational overheads over the standard RS.

## 2  MEMORABLE INFORMATION CRITERION

Firstly we formalize the online memory selection problem and propose our information-theoretic criteria to detect informative points. Then we introduce a Bayesian model to compute the proposed criteria efficiently. Finally, we demonstrate the usefulness of the criteria in a greedy algorithm.

### 2.1  ONLINE MEMORY SELECTION

We consider a supervised learning problem, where the agent evolves a predictor $f$ to learn the underlying mapping from inputs to targets. We assume that the predictor is in the following form,

$$f_{\boldsymbol{\theta}}(\cdot) = g_{\boldsymbol{\theta}_g}(h_{\boldsymbol{\theta}_h}(\cdot)), \tag{1}$$

where $\boldsymbol{\theta} := (\boldsymbol{\theta}_g, \boldsymbol{\theta}_h)$. The base network $h_{\boldsymbol{\theta}_h}(\mathbf{x})$ extracts the feature representation of the input $\mathbf{x}$, and $g_{\boldsymbol{\theta}_g}$ maps the representation to predictions. The function $g_{\boldsymbol{\theta}_g}$ is usually a simple module, such as a linear layer (Krizhevsky et al., 2012) or mean-of-neighbours (Snell et al., 2017) mapping to the softmax logits. We assume the memory buffer contains the inputs, features, predictions, and targets,

$$\mathcal{M} = \{(\mathbf{x}_m, \mathbf{h}_m, \mathbf{g}_m, y_m)\}_{m=1}^M . \tag{2}$$

In the online memory selection problem, the agent iteratively receives observations from an online stream, and updates the predictor and the memory buffer. Let $\mathcal{B} = \{(\mathbf{x}_b, y_b)\}_{b=1}^B$ be the observed batch in one iteration, a general update formula can be expressed as,

$$(\boldsymbol{\theta}, \mathcal{M}) \longleftarrow (\boldsymbol{\theta}, \mathcal{M}, \mathcal{B}) . \tag{3}$$

We emphasize that updating the predictor parameters $\boldsymbol{\theta}$ and updating the memory $\mathcal{M}$ are essential but orthogonal components. For example, the agent can update the memory using RS (Buzzega et al., 2020) or GSS (Aljundi et al., 2019b), and update $\boldsymbol{\theta}$ using GEM (Lopez-Paz & Ranzato, 2017) or experience replay (Chaudhry et al., 2019b). This section focuses on how to maintain a representative memory $\mathcal{M}$ online, while how to update the predictor $f_{\boldsymbol{\theta}}$ is not our focus. Furthermore, unlike task-based continual learning (Kirkpatrick et al., 2017), we assume the agent receives only the data stream without observing task boundaries. This scenario is more difficult for memory selection. For instance, it must deal with data imbalances, e.g., if discrete tasks of varying lengths exist, we may want the selection to identify a more balanced memory and avoid catastrophic forgetting.

### 2.2  MEMORABLE INFORMATION CRITERION

Intuitively, the online memory should maintain points that carry useful information. We consider a Bayesian model for this problem since the Bayesian framework provides a principled way to quantify the informativeness of data points. Specifically, we let $p(y|\mathbf{w}; \mathbf{x})$ be the likelihood of the observation $y$ given the deterministic input $\mathbf{x}$ dependent on the random model parameter $\mathbf{w}$,[1] and we

---

[1]The density $p$ and the parameter $\mathbf{w}$ here are left completely generic.

denote by $p(\mathbf{w})$ the prior distribution. Then the posterior of $\mathbf{w}$ and the predictive distribution of new data points can be inferred according to Bayes's rule. Also worth mentioning, the Bayesian model is introduced merely for memory selection, which is different from the predictor $f_{\boldsymbol{\theta}}$.

In each iteration of online memory selection, the agent decides whether to include the new data point $(\mathbf{x}_\star, y_\star)$ into the memory $\mathcal{M}$. To this end, we present information-theoretic criteria based on the Bayesian model to evaluate the usefulness of the new data point. For notational simplicity, we denote by $\mathbf{X}_\mathcal{M}$ and $\mathbf{y}_\mathcal{M}$ the inputs and the targets in the memory, respectively.

**Surprise.** As Claude Shannon says: "Information is the resolution of uncertainty". If the memory $\mathcal{M}$ is certain of the target $y_\star$ of $\mathbf{x}_\star$ beforehand, incorporating the data point $(\mathbf{x}_\star, y_\star)$ brings little information into $\mathcal{M}$. Thus we propose to measure the "surprise" of the new point given the memory. We define the *surprise* criterion of $(\mathbf{x}_\star, y_\star)$ given $\mathcal{M}$ as the negative log conditional probability,

$$s_{\mathrm{surp}}((\mathbf{x}_\star, y_\star); \mathcal{M}) = -\log p(y_\star|\mathbf{y}_\mathcal{M}; \mathbf{X}_\mathcal{M}, \mathbf{x}_\star), \tag{4}$$

where $p(y_\star|\mathbf{y}_\mathcal{M}; \mathbf{X}_\mathcal{M}, \mathbf{x}_\star) = \int p(y_\star|\mathbf{w}; \mathbf{x}_\star)p(\mathbf{w}|\mathbf{y}_\mathcal{M}; \mathbf{X}_\mathcal{M})\mathrm{d}\mathbf{w}$. Intuitively, if $(\mathbf{x}_\star, y_\star)$ comes from a new task, it will likely be less predictable according to the memory and the surprise will be large. In this way, the surprise criterion detects informative points and ensures diversity within the buffer.

**Learnability.** The *surprise* detects unpredictable points in the data stream, but it is not capable of distinguishing "helpful" unfamiliar points and "harmful" outliers. To avoid the outliers, we propose the *learnability* criterion, which measures how much our Bayesian model can explain the new point once it has absorbed its information in the memory. Formally, we define the *learnability* as follows,

$$s_{\mathrm{learn}}((\mathbf{x}_\star, y_\star); \mathcal{M}) = \log p(y'_\star|y_\star, \mathbf{y}_\mathcal{M}; \mathbf{X}_\mathcal{M}, \mathbf{x}_\star)|_{y'_\star = y_\star}, \tag{5}$$

where we use $y'_\star$ and $y_\star$ to represent two realizations of the same random variable. Intuitively, the learnability measures the consensus between the new point and the memory. We illustrate the utility of learnability through a toy example in Figure 1. We use a Gaussian process with the RBF kernel to fit memory points (shown as black dots ●). We visualize its pre-

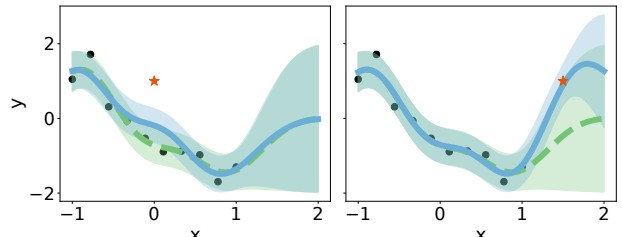

Figure 1: Visualization of the toy problem.

dictive mean and $95\%$ confidence interval as the green line and the shadow region. For two new points (shown as red stars ⋆) separately in the left and right figure, we also visualize the predictive distribution given the memory and the new point, shown as the blue line and the shadow region. We observe that both points are surprising to the memory since they lie outside (or just outside) of the green confidence regions. However, after conditioning on the new point itself, the left point still lies outside of the confidence region while the prediction on the right adjusts to include the new point. Thus the left point has low learnability, and the right point has high learnability, though both have high surprises. Thus the learnability can be used to detect outliers when combined with the surprise.

**Memorable Information Criterion (MIC).** Given $\eta \geq 0$, we propose the Memorable Information Criterion (MIC) as a combination of the surprise and learnability of the new point,

$$\mathrm{MIC}_\eta((\mathbf{x}_\star, y_\star); \mathcal{M}) = \eta s_{\mathrm{learn}}((\mathbf{x}_\star, y_\star); \mathcal{M}) + s_{\mathrm{surp}}((\mathbf{x}_\star, y_\star); \mathcal{M}). \tag{6}$$

Notably, the MIC is related to the information gain (IG) (Cover, 1999): $\mathrm{IG}((\mathbf{x}_\star, y_\star); \mathcal{M}) = \mathrm{KL}\left(p(\mathbf{w}|y_\star, \mathbf{y}_\mathcal{M}; \mathbf{X}_\mathcal{M}, \mathbf{x}_\star)||p(\mathbf{w}|\mathbf{y}_\mathcal{M}; \mathbf{X}_\mathcal{M})\right)$. By Appendix E, IG can be rewritten as,

$$\mathrm{IG}((\mathbf{x}_\star, y_\star); \mathcal{M}) = \mathbb{E}_{p(\mathbf{w}|y_\star, \mathbf{y}_\mathcal{M}; \mathbf{X}_\mathcal{M}, \mathbf{x}_\star)}\left[\log p(y_\star|\mathbf{w}; \mathbf{x}_\star)\right] - \log p(y_\star|\mathbf{y}_\mathcal{M}; \mathbf{X}_\mathcal{M}), \tag{7}$$

where the second term is the surprise, and the first term is upper bounded by the learnability based on Jensen's inequality. Thus we have that $\mathrm{MIC}_1((\mathbf{x}_\star, y_\star); \mathcal{M}) \geq \mathrm{IG}((\mathbf{x}_\star, y_\star); \mathcal{M}) \geq 0$. Given the connection, we generalize IG to be a weighted combination of learnability and surprise as well in the Appendix. We also compare with an entropy reduction criterion. More discussions are in Sec C.

## 2.3 An Efficient Bayesian Model

Computing the MIC requires inferring Bayesian posteriors, which can be computationally unfeasible for complicated Bayesian models. Now we combine deep networks and Bayesian linear models to make computing the MIC efficient. Deep networks are well-known for learning meaningful

representations based on which the predictions are made by simple modules such as linear layers (Krizhevsky et al., 2012; Dubois et al., 2021). Thus we treat our base network $h_{\boldsymbol{\theta}_h}$ as a feature extractor and establish the Bayesian model from the feature space to the target space. Precisely, let $d_0$ be the feature dimension and $\mathbf{h}_0 = h_{\boldsymbol{\theta}_h}(\mathbf{x}) \in \mathbb{R}^{d_0}$ be the feature of the input $\mathbf{x}$, we compute the normalized feature $\mathbf{h} := [\mathbf{h}_0^\top, 1]^\top / \sqrt{d}$ for $d := d_0 + 1$. Then we consider a Bayesian linear model,

$$y = \mathbf{w}^\top \mathbf{h} + \epsilon, \epsilon \sim \mathcal{N}(0, \sigma^2) . \tag{8}$$

We consider an isotropic Gaussian prior $\mathbf{w} \sim \mathcal{N}(0, \sigma_w^2 \mathbf{I})$. Denoting the features and the targets in the buffer as $\mathbf{H}_{\mathcal{M}} \in \mathbb{R}^{M \times d}$ and $\mathbf{y}_{\mathcal{M}} \in \mathbb{R}^M$ respectively, the weight posterior can be computed as,

$$p(\mathbf{w}|\mathbf{y}_{\mathcal{M}}; \mathbf{H}_{\mathcal{M}}) = \mathcal{N}(\mathbf{A}_{\mathcal{M}}^{-1} \mathbf{b}_{\mathcal{M}}, \sigma^2 \mathbf{A}_{\mathcal{M}}^{-1}), \tag{9}$$

where we define $\mathbf{A}_{\mathcal{M}}^{-1} = \left( \mathbf{H}_{\mathcal{M}}^\top \mathbf{H}_{\mathcal{M}} + c\mathbf{I}_d \right)^{-1}$ for $c = \frac{\sigma^2}{\sigma_w^2}$ and $\mathbf{b}_{\mathcal{M}} = \mathbf{H}_{\mathcal{M}}^\top \mathbf{y}_{\mathcal{M}}$. Given a new point $(\mathbf{x}_\star, y_\star)$, let $\mathbf{h}_\star$ be the corresponding feature, then the MIC can be computed explicitly as,

$$\begin{aligned} \text{MIC}_\eta((\mathbf{x}_\star, y_\star); \mathcal{M}) = &\eta \log \mathcal{N}(y_\star | \mathbf{h}_\star^\top \mathbf{A}_{\mathcal{M}+}^{-1} \mathbf{b}_{\mathcal{M}+}, \sigma^2 \mathbf{h}_\star^\top \mathbf{A}_{\mathcal{M}+}^{-1} \mathbf{h}_\star + \sigma^2) \\ &- \log \mathcal{N}(y_\star | \mathbf{h}_\star^\top \mathbf{A}_{\mathcal{M}}^{-1} \mathbf{b}_{\mathcal{M}}, \sigma^2 \mathbf{h}_\star^\top \mathbf{A}_{\mathcal{M}}^{-1} \mathbf{h}_\star + \sigma^2), \end{aligned} \tag{10}$$

where $\mathbf{A}_{\mathcal{M}+} = \mathbf{A}_{\mathcal{M}} + \mathbf{h}_\star \mathbf{h}_\star^\top$ and $\mathbf{b}_{\mathcal{M}+} = \mathbf{b}_{\mathcal{M}} + \mathbf{h}_\star y_\star$. A derivation is provided in Appendix E.

**Efficient computation.** Computing the MIC still involves the matrix inversions $\mathbf{A}_{\mathcal{M}+}^{-1}$ and $\mathbf{A}_{\mathcal{M}}^{-1}$. Fortunately, since $\mathbf{A}_{\mathcal{M}+}$ differ with $\mathbf{A}_{\mathcal{M}}$ by a rank-one matrix, its inverse can be computed efficiently given $\mathbf{A}_{\mathcal{M}}^{-1}$ according to the Sherman-Morrison Formula (Sherman & Morrison, 1950),

$$\mathbf{A}_{\mathcal{M}+}^{-1} = \mathbf{A}_{\mathcal{M}}^{-1} - \frac{\mathbf{A}_{\mathcal{M}}^{-1} \mathbf{h}_\star \mathbf{h}_\star^\top \mathbf{A}_{\mathcal{M}}^{-1}}{1 + \mathbf{h}_\star^\top \mathbf{A}_{\mathcal{M}}^{-1} \mathbf{h}_\star}. \tag{11}$$

Moreover, $\mathbf{A}_{\mathcal{M}+}^{-1}$ needs not to be computed explicitly either since the MIC expression involves only the matrix vector product $\mathbf{A}_{\mathcal{M}+}^{-1} \mathbf{h}_\star$. As a result, given $\mathbf{A}_{\mathcal{M}}^{-1}$ and $\mathbf{b}_{\mathcal{M}}$, the computational cost of the MIC is merely $\mathcal{O}(d^2)$. One can store $\mathbf{A}_{\mathcal{M}}^{-1}$ and $\mathbf{b}_{\mathcal{M}}$ along with the memory, so that the MIC can be computed efficiently. Finally, since the buffer adds or removes at most one point in each iteration, $\mathbf{A}_{\mathcal{M}}^{-1}$ and $\mathbf{b}_{\mathcal{M}}$ can be updated efficiently in $\mathcal{O}(d^2)$ as well using the Sherman-Morrison formula.

**Feature update.** Since the predictor evolves along with memory selection, the feature of each input also changes. The feature $h_\theta(\mathbf{x}_\star)$ is usually available since the agent needs to forward $\mathbf{x}_\star$ for learning the predictor. However, the stored features $\mathbf{H}_{\mathcal{M}}$ in the buffer are usually outdated due to representation shifting in the online process. Forwarding the whole memory through the network in each iteration is also computationally infeasible. Alternatively, we notice that the memory is usually used to aid learning in the online process, such as memory replay in continual learning (Rebuffi et al., 2017) or experience replay in deep Q learning (Mnih et al., 2015). Therefore, we update the memory features whenever the agent forwards the memories through the network,[2] which is free of computational overheads. Finally, we note that the matrix $\mathbf{A}_{\mathcal{M}}^{-1}$ needs to be recomputed when the memory features are updated, which takes $\mathcal{O}(d^3)$ computations using a Cholesky decomposition.

**Classification.** For classification problems, the exact Bayesian posterior is no longer tractable, preventing explicit expressions of the MIC. To alleviate this problem, we propose to approximate the classification problem as a multi-output regression problem. Specifically, let $1 \le y \le K$ be the label among $K$ classes, we use the Bayesian model to regress the one-hot vector $\hat{y} \in \mathbb{R}^K$, where $\hat{y}_i = \delta_{iy}, i = 1, .., K$. Further, we let our Gaussian prior to be independent across output dimensions. Then, the MIC can be computed as the summation of MICs across each output dimension.

## 2.4 Demonstrating the Criteria By A Greedy Algorithm

Using a greedy algorithm for online memory selection, we show that the proposed information-theoretic criteria encourage informative points. Intuitively, in each iteration, the algorithm adds the most informative new point and removes the least informative existing point in the buffer.

**Information criterion for memories.** Consider the new data $(\mathbf{x}_\star, y_\star)$, the buffer $\mathcal{M}$, and a data point $(\mathbf{x}_m, y_m) \in \mathcal{M}$. Let $\mathcal{M}_{\star, -m} := \mathcal{M} \cup (\mathbf{x}_\star, y_\star) \backslash (\mathbf{x}_m, y_m)$. We measure the informativeness

---

[2]The features are updated in every iteration in our continual learning experiments, due to the memory replay.

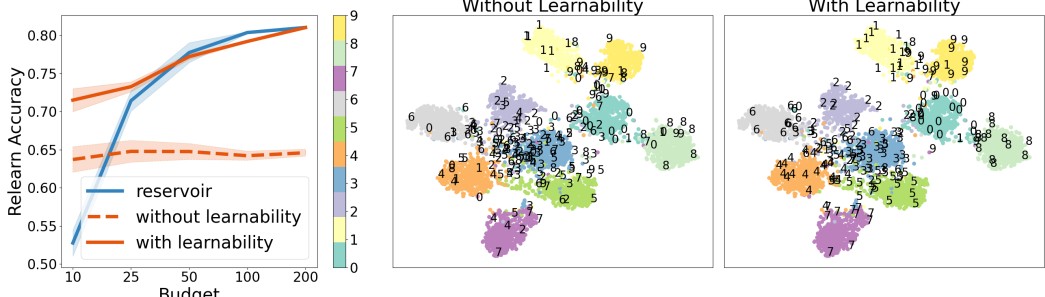

Figure 2: *left*: Relearning performance of the memory. We compare InfoGS with RS and InfoGS without using learnability, for various budgets. *middle, right*: TSNE visualizations of the training data (colored dots, where the color represents the label of the data point) and 200 memory points (black digits, where the digit represents the label of the point) for InfoGS without and with learnability, respectively. We observe that InfoGS without learnability selects many memories that differ from the training majority, i.e., it selects outliers. In comparison, incorporating learnability avoids the outliers and selects memories that spread the training region and help to learn the decision boundary. Further, InfoGS performances also improve over RS when the budget is very low.

of $(\mathbf{x}_m, y_m)$ according to its MIC for the pseudo buffer $\mathcal{M}_{\star, -m}$, i.e., $\mathrm{MIC}_\eta((\mathbf{x}_m, y_m); \mathcal{M}_{\star, -m})$. Notably, the criterion for memories is comparable to the criterion for the new point, since $\mathrm{MIC}_\eta((\mathbf{x}_\star, y_\star); \mathcal{M}) = \mathrm{MIC}_\eta((\mathbf{x}_\star, y_\star); \mathcal{M}_{\star, -\star})$ can be expressed similarly.

Let $(\mathbf{x}_{b^\star}, y_{b^\star}) \in \mathcal{M}$ be the memory point with the smallest MIC. We decide whether to replace it with the new point $(\mathbf{x}_\star, y_\star)$. Specifically, we present *information improvement thresholding* to determine whether the new point improves the information by at least a certain amount. We present *learnability thresholding* to only add points with high learnability. The overall algorithm, Information-theoretic Greedy Selection (InfoGS), replaces $(\mathbf{x}_{b^\star}, y_{b^\star})$ with $(\mathbf{x}_\star, y_\star)$, if the new point passes both thresholding. A detailed description and the pseudocode are shown in Alg 2 in the Appendix.

**A toy experiment.** Using InfoGS, we illustrate the proposed criteria by a toy experiment. We generate the toy dataset by training a ResNet-18 (He et al., 2016) for CIFAR-10 classification and storing the network features and the targets of the whole dataset. We use pre-trained features for InfoGS, to focus on the online memory selection without the interference of feature learning. The agent observes the dataset stream for one epoch with the batch size 32, sequentially by the order of classes: $\{0, 1\}, \{2, 3\}, \{4, 5\}, \{6, 7\}, \{8, 9\}$. We evaluate the final memory by the "relearn" performance, where we fit a linear classifier on merely the memory points and report the test accuracy. In this way, the relearning performance directly represents the quality of the memory. We compare InfoGS with RS. To investigate the impact of learnability, we also include a modified InfoGS by removing the learnability. We run all methods across various memory budgets. We also plot the memories using the TSNE visualization (Hinton & Roweis, 2002). The results are shown in Figure 2.

## 3 INFORMATION-THEORETIC RESERVOIR SAMPLING

In this section we incorporate the MIC into continual learning. For training the prediction network, we pick the dark experience replay (DER++) (Buzzega et al., 2020), which is fully compatible to GCL and achieves state-of-art performances on continual learning benchmarks. DER++ prevents the network from forgetting by regularizing both the targets and the logits. Specifically, for each new batch $\mathcal{B}$, DER++ optimizes the network parameter $\boldsymbol{\theta}$ by minimizing the following objective,

$$\mathcal{L}(\boldsymbol{\theta}; \mathcal{M}) = \frac{1}{|\mathcal{B}|} l(\mathcal{B}; \boldsymbol{\theta}) + \frac{\alpha}{M} \sum_{m=1}^{M} \|f_{\boldsymbol{\theta}}(\mathbf{x}_m) - \mathbf{g}_m\|_2^2 + \frac{\beta}{M} \sum_{m=1}^{M} l((\mathbf{x}_m, y_m); \boldsymbol{\theta}),$$

where the objective is composed of (1) the fitting loss of the current batch, such as the cross entropy; (2) the mean squared loss between the predictive logits $f_{\boldsymbol{\theta}}(\mathbf{x}_m)$ of the memory input $\mathbf{x}_m$ and its memory logits $\mathbf{g}_m$; (3) the fitting loss of the memory. $\alpha$ and $\beta$ are hyper-parameters.

While straightforward, naively employing InfoGS to curate the buffer suffers the timing issue we identify for continual learning. We present a new algorithm, the information-theoretic reservoir sampling, which integrates the MIC into RS and achieves improved robustness against data imbalance.

---

**Algorithm 1** Information-theoretic Reservoir Sampling (InfoRS)

---

1: **Input:** Memory $\mathcal{M}$ and matrices $\mathbf{A}_{\mathcal{M}}^{-1}, \mathbf{b}_{\mathcal{M}}$, the batch $\mathcal{B}$, the predictor parameter $\boldsymbol{\theta}$.
2: **Input:** The reservoir count $n$ and the budget $M$.
3: **Input:** Running mean and stddev for the MIC: $\hat{\mu}_i, \hat{\sigma}_i$. The thresholding ratio $\gamma_i$.
4: Update the predictor parameter $\boldsymbol{\theta}$ based on $\mathcal{M}$ and $\mathcal{B}$.[3]                       // Predictor Update
5: Update the features for the memory points used in replay, and update $\mathbf{A}_{\mathcal{M}}^{-1}, \mathbf{b}_{\mathcal{M}}$ accordingly.
6: **for** $(\mathbf{x}_\star, y_\star)$ in $\mathcal{B}$ **do**
7:     **if** $|\mathcal{M}| < M$ **or** $\mathrm{MIC}_\eta((\mathbf{x}_\star, y_\star); \mathcal{M}) \geq \hat{\mu}_i + \hat{\sigma}_i * \gamma_i$          // Information Thresholding
8:         Update $\mathcal{M}, n \longleftarrow$ **ReservoirSampling**$(\mathcal{M}, M, n, (\mathbf{x}_\star, y_\star))$.         // Memory Update
9:         Update $\mathbf{A}_{\mathcal{M}}^{-1}, \mathbf{b}_{\mathcal{M}}$ based on the Sherman-Morrison formula if $\mathcal{M}$ is updated.
10:     Update $\hat{\mu}_i, \hat{\sigma}_i$ using the criterion $\mathrm{MIC}_\eta((\mathbf{x}_\star, y_\star); \mathcal{M})$.         // Running Moments Update
11: **return** Buffer $\mathcal{M}$ and $\mathbf{A}_{\mathcal{M}}^{-1}, \mathbf{b}_{\mathcal{M}}$. The reservoir count $n$ and statistics $\hat{\mu}_i, \hat{\sigma}_i$. The updated $\boldsymbol{\theta}$.

---

### 3.1 THE IMPORTANCE OF THE TIMING TO UPDATE THE MEMORY

While maintaining a representative memory is critical to properly regularize an online CL process, we will now discuss and empirically show that *when* the memory is updated (i.e., the update timing) is also influential. We begin by giving an intuition of this issue with a mind experiment, where the agent sequentially observes the data from two tasks $\mathcal{T}_1$ and $\mathcal{T}_2$. At the time $t$ of task switching, the agent has a memory with $M$ examples from $\mathcal{T}_1$. Assume that the agent will update the memory by removing $\frac{M}{2}$ examples from $\mathcal{T}_1$ and adding $\frac{M}{2}$ examples from $\mathcal{T}_2$ at the time $t + \Delta t$. Since the $\mathcal{T}_2$ examples are repeatedly visited after time $t$, it is favorable to use a large $\Delta t$. In contrast, updating the memory at the beginning of $\mathcal{T}_2$ with $\Delta t = 0$, leads to degrading $\mathcal{T}_1$ regularizations too early.

We further demonstrate this by a Split CI-FAR10 experiment with 5 tasks. After $\Delta t$ iterations of each task $\mathcal{T}_i$ for $i = 1, ..., 5$, we randomly replace $\frac{200}{i}$ memories with the examples from $\mathcal{T}_i$. In Figure 3, we report the final accuracies with varying $\Delta t$ to change the timing of memory updates, for both $\alpha = 0, \beta = 1$ and $\alpha = 0.3, \beta = 1$ in dark experience replay. We observe that updating the memory early in each task leads to worse performances. The phenomenon is more distinct for $\alpha = 0.3$, where storing premature logits $\mathbf{g}_m$ early in each task actually hurts the network learning.

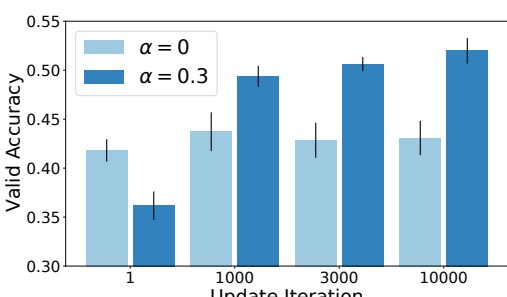

Figure 3: The effect of update timing $\Delta t$.

Reservoir sampling acts randomly to maintain a uniform sample from the past stream, thus it tends not to update the buffer immediately at a new task. However, InfoGS accumulates surprising examples greedily, thus its memory updates tend to happen early in each task. Therefore, the greediness of InfoGS is a disadvantage to RS in terms of the timing to update the memory. Moreover, the importance of timing persists not only in the task-based continual learning, but also in the general continual learning regime where the data distribution shifts continuously.

### 3.2 INFORMATION-THEORETIC RESERVOIR SAMPLING

To alleviate the timing problem of InfoGS, we present the information-theoretic reservoir sampling (InfoRS) algorithm, which attempts to bring together the merits of both the information-theoretic criteria and the reservoir sampling. The MIC facilitates a representative memory, but the greedy selection should be circumvented for the sake of timing. To this end, we propose InfoRS, which conducts reservoir sampling over those points with high MICs. Similarly to InfoGS, we let $\hat{\mu}_i, \hat{\sigma}_i$ be the running mean and standard deviation of the MICs for all historical observations, and let $\gamma_i$ be a hyper-parameter. InfoRS selects the point $(\mathbf{x}_\star, y_\star)$ for reservoir sampling if its MIC passes the *information threshold*: $\mathrm{MIC}_\eta((\mathbf{x}_\star, y_\star); \mathcal{M}) \geq \hat{\mu}_i + \hat{\sigma}_i * \gamma_i$. The overall algorithm of InfoRS is shown in Alg 1 and we also provide the algorithm of ReservoirSampling in Alg 3 in the Appendix.

---

[3] We use DER++ in Eq 12 for updating the predictor, but other algorithms can be adopted as well.

Worth mentioning, a point $(\mathbf{x}_\star, y_\star)$ passing the information threshold results in itself being considered in the reservoir sampling procedure. However, due to the stochasticity in RS, the point might not be added to the memory. Consequently, InfoRS overcomes the timing issue for the greedy InfoGS. In addition, standard RS draws uniform samples over the stream, making it vulnerable to imbalanced data. For example, the memory can be dominated by a specific task if it has lots of observations, while examples from other tasks might be missed and their regularizations be degraded. In comparison, for InfoRS, if a specific task occupies more space in the memory, similar points in the task will have a smaller information criterion and thus a lower chance to pass the information threshold. Thus the memory selected by InfoRS will not be dominated by specific tasks and tend to be more balanced. In this way, InfoRS could be more robust to data imbalance compared to RS.

## 4 RELATED WORKS

**Experience replay for continual learning.** Experience replay is widely adopted in continual learning to prevent catastrophic forgetting (Robins, 1995; Lopez-Paz & Ranzato, 2017; Rebuffi et al., 2017; Rolnick et al., 2019). However, the majority of approaches focus on task-based continual learning and update the memory relying on task boundaries (Rebuffi et al., 2017; Nguyen et al., 2018; Titsias et al., 2020; Pan et al., 2020; Bang et al., 2021; Yoon et al., 2021). In contrast, reservoir sampling (Vitter, 1985) is used (Rolnick et al., 2019; Chaudhry et al., 2019b; Buzzega et al., 2020; Isele & Cosgun, 2018; Balaji et al., 2020) to update the memory online and maintain uniform samples from the stream. RS is generalized to class-balanced RS (CBRS) that selects uniform samples over each class (Chrysakis & Moens, 2020; Wiewel & Yang, 2021; Kim et al., 2020), but CBRS cannot resolve other forms of data imbalance. To counter the vulnerabilities to data imbalance, GSS (Aljundi et al., 2019b) proposes to minimize gradient similarities to ensure the memory diversity, but incurs large computational costs for gradient computations. For continual reinforcement learning, Isele & Cosgun (2018) propose selection strategies based on prediction errors or rewards, but these do not promote diversities within the memory. They further attempt to maximize the coverage of the input space for the memory, while selecting an appropriate distance metric between high-dimensional inputs is crucial. Borsos et al. (2020) present the weighted coreset selection and extend it to GCL. However, the method incurs large computational costs and is limited to a small memory.

**More applications with online data selection.** Apart from preventing catastrophic forgetting, online selecting data is also useful for improving the computational and memory efficiencies of learning systems. Deep Q-learning (Mnih et al., 2015) adopts past experiences to train the Q-networks, and selective experiences can improve the learning efficiency of agents (Schulman et al., 2015). Moreover, online selection forms the key component in curriculum learning (Bengio et al., 2009) to pick the next curriculum for efficient learning. Specifically, prediction gain (PG) (Bellemare et al., 2016; Graves et al., 2017) picks the curriculum according to $\mathcal{L}((\mathbf{x}_\star, y_\star); \boldsymbol{\theta}) - \mathcal{L}((\mathbf{x}_\star, y_\star); \boldsymbol{\theta}')$, where $\boldsymbol{\theta}'$ is the parameter after the network updates $\boldsymbol{\theta}$ for one step using $(\mathbf{x}_\star, y_\star)$. Thus PG can be seen as the combination of surprises and learnability as well. However, PG does not use a memory buffer and the $\boldsymbol{\theta}'$ is obtained by one-step gradient updates unlike our exact inference. Bayesian optimization (Mockus et al., 1978) and active learning (MacKay, 1992) learn the problem with querying the training data interactively to maximize some unknown objective or data efficiency. However, the labels of the queried data are not available before the query, which is different to our setting where both the input and target are available. Online submodular maximization (Buchbinder et al., 2014; Lavania et al., 2021) selects a memory buffer online to maximize a submodular criterion. They propose greedy algorithms based on an improvement thresholding procedure, which is similar to our InfoGS. However, as discussed in Sec. 3, greedy algorithms can suffer from the issue of timing in continual learning. Also, while we provide MIC as a general criterion to evaluate forthcoming points, submodular maximization relies on meaningful submodular criterions.

**Offline data selection.** Offline selections assume the whole dataset is available and a representative small set is targeted, which helps to lift the memory burden of dataset storage and improve learning efficiency. Coreset selection (Campbell & Broderick, 2019) aims to find a weighted set whose maximum likelihood estimator approximates that of the whole dataset. Dataset distillation (Zhao et al., 2020; Nguyen et al., 2021) aims at a small set training on which achieves similar performances to training on the original dataset. Sparse Gaussian processes also consider the selection of inducing points (Seeger et al., 2003; Csato & Opper, 2002; Bui et al., 2017). For instance, Seeger et al. (2003) adopt the information gain to select inducing points for a scalable Gaussian process method.

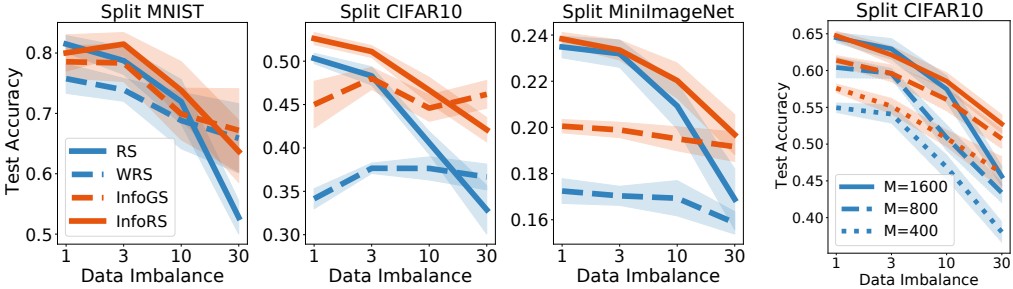

Figure 4: Test performances against data imbalance over continual learning benchmarks. *left three:* We compare RS, weighted RS (WRS) using the output-space Hessian, InfoGS and InfoRS. For each experiment, we plot the mean and the $95\%$ confidence interval across 10 random seeds. *right:* We compare InfoRS and RS for different memory budgets over Split CIFAR10. For all figures, we observe that InfoRS is more robust to data imbalance compared to RS.

## 5 EXPERIMENTS

We investigate how InfoRS performs empirically. We assume the general continual learning regime, where the agent is agnostic of task boundaries. We use DER++ (Buzzega et al., 2020) for learning the predictor along with the online memory selection instead of using pretrained features. To introduce experimental details, we first describe useful notations following Buzzega et al. (2020). We denote the batch size as *bs*, and the memory batch size as *mbs*. Specifically, we randomly sample *mbs* points from the memory buffer in each iteration. We denote the memory budget as $M$. For all experiments, we adopt the stochastic gradient descent optimizer and we denote the learning rate as *lr*.

**Benchmarks.** The benchmarks involve Permuted MNIST, Split MNIST, Split CIFAR10, and Split MiniImageNet. Permuted MNIST involves 20 tasks, and transforms the images by task-dependent permutations. The agent attempts to classify the digits without task identities. The other benchmarks split the dataset into disjoint tasks based on labels. For Split MNIST and Split CIFAR10, 10 classes are split into 5 tasks evenly; for Split MiniImageNet, 100 classes are split into 10 tasks evenly. We use a fixed split across all random seeds. The agent attempts to classify the images without task identities either. For the sake of space, we present the results for Permuted MNIST in the Appendix.

**Methods.** The online fashion in GCL causes that only a few existing methods are compatible. In addition to RS, weighted reservoir sampling (WRS) (Chao, 1982; Efraimidis & Spirakis, 2006) generalizes RS to non-uniform sampling in proportional to data-dependent weights. Consequently, one can adopt WRS to favor specific points according to the chosen importance weight. We include a WRS baseline with the total output-space Hessian (Pan et al., 2020) as importance weights. Specifically, if $p_{1:K}$ are the predictive probabilities for $K$ classes of a point, its total output-space Hessian is computed as $1 - \sum_{k=1}^{K} p_i^2$. Intuitively, the total Hessian of a point is small if it has a confident prediction. We compare InfoRS, InfoGS, RS and WRS over the experiments. Additionally, class-balanced reservoir sampling (CBRS) (Chrysakis & Moens, 2020) maintains uniform samples separately for each class, and GSS (Aljundi et al., 2019b) minimizes the gradient similarities in the memory. We present the comparisons with CBRS and GSS in Figure 8 in the Appendix for clarity.

**Experimental details.** For Permuted MNIST and Split MNIST, we adopt a fully connected network with two hidden layers of 100 units each, and we set M=100, *bs*=128, *mbs*=128. For Split CIFAR10 and Split MiniImageNet, we adopt a standard ResNet-18 (He et al., 2016) and set *bs*=32, *mbs*=32.[4] We set $M = 200$ and $M = 1000$ for Split CIFAR10 and Split MiniImageNet, respectively. To tune the hyper-parameters, we pick $10\%$ of training data as the validation set, then we pick the best hyper-parameter based on the averaged validation accuracy over 5 random seeds. The final test performance is reported by relearning the model using the whole training set with the best hyper-parameter, averaged over 10 seeds. The tuning hyper-parameters include the learning rate *lr*, the logit regularization coefficient $\alpha$, the target regularization coefficient $\beta$, the learnability ratio $\eta$, and the information thresholding ratio $\gamma_i$, if needed. We present the detailed hyper-parameters in Table 1.

---

[4]Our ResNet contains a MaxPooling with window shape 3 and stride 2 after the first ReLU, which is absent in Buzzega et al. (2020). Our initial conv uses kernel 7 and stride 2, compared to their kernel 3 and stride 1.

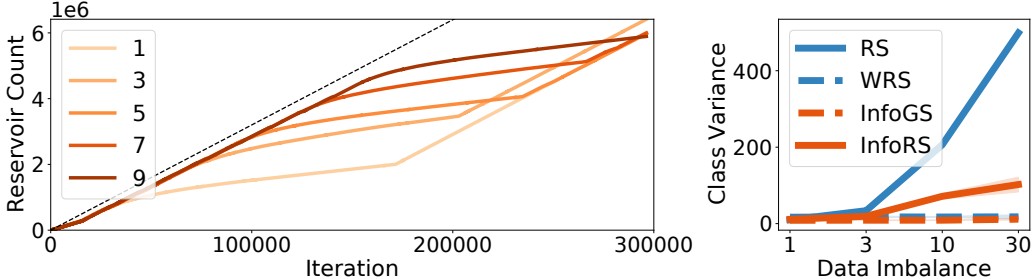

Figure 5: *left*: The reservoir count along with training. The line $i = 1, 3, 5, 7, 9$, corresponds to the problem where the $i$-th task has 10 times epochs than the other tasks. We also plot the black dashed line for the count of standard RS. We observe that the reservoir count of InfoRS grows at a similar speed as RS for most tasks. However, it grows slower at iterations corresponding to task $i$ for each line $i$. In this way, InfoRS counters the imbalanced data stream, maintains a diverse buffer, and behaves more robustly. *right*: The class variance of the memory. We observe that the memory of RS becomes the most imbalanced with the data imbalance increasing.

**Robustness against data imbalance.** To simulate data imbalance, we vary training epochs across different tasks. Specifically, given the base epoch $n_e > 0$ and data imbalance $r \geq 1$, we randomly pick one task $\mathcal{T}_\star$ and train it for $n_e * r$ epochs, while we train the other tasks for $n_e$ epochs. Thus $\mathcal{T}_\star$ is observed $r$ times more frequently and the data imbalance increases when $r$ grows. Given the imbalanced training data, the agent is still expected to learn all tasks well. Therefore, we evaluate all tasks equally and report the averaged test performance. For all the benchmarks, we conduct experiments with $r = 1, 3, 10, 30$. We set $n_e = 1$ for Permuted MNIST and Split MNIST, $n_e = 50$ for Split CIFAR10, and $n_e = 100$ for Split MiniImageNet. The results are shown in Figure 4. Moreover, Figure 4 also compares InfoRS and RS across varying budgets over Split CIFAR10.

We observe that RS and InfoRS outperform other approaches when the data is balanced, while InfoRS demonstrates improved robustness compared to RS as data imbalance increases. Compared to InfoRS and RS, InfoGS is less affected by data imbalance since InfoGS acts greedily to gather the memories. Also, WRS with the total Hessian is more robust than RS, since frequently observed points tend to have confident predictions and thus small weights to be sampled. Overall, we observe that InfoRS improves the robustness from RS without sacrificing the performance for balanced data.

We manifest the robustness of InfoRS by visualizing its reservoir count along with training, which is the total number of points considered in the reservoir sampling. Thus the count increases linearly for the standard RS, but is affected by the information thresholding in InfoRS. We track the reservoir count of InfoRS for Split MiniImageNet with data imbalance $r = 10$. The results are visualized in Figure 5. Furthermore, a good memory should contain similar example amounts for all classes. Let $M_{1:K}$ be the example numbers in the memory for all $K$ classes. For each method, we compute the variance $\frac{1}{K} \sum_k M_k^2 - (\frac{1}{K} \sum_k M_k)^2$. Figure 5 also visualizes the variances for Split MiniImageNet.

**Running time.** To demonstrate the computational efficiency of the proposed approaches, we compare the running time of RS, WRS, InfoGS, InfoRS. Specifically, we plot the total running time over Split MiniImageNet with the data imbalance $r = 30$. As shown in Figure 6, InfoGS and InfoRS incur small computational overheads compared to RS. This result demonstrates the efficiency of the proposed algorithms.

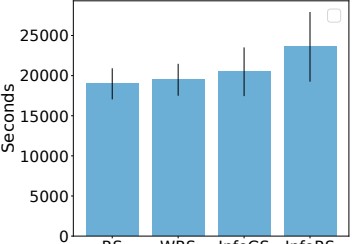

Figure 6: The total running time.

## 6 CONCLUSION

We presented information-theoretic approaches for online memory selection. Specifically, we proposed to maintain the most information in the memory by evaluating new points along two dimensions: the *surprise* and the *learnability*, which are shown to select informative points and avoid outliers. Besides the InfoGS which updates the buffer whenever seeing new examples, we presented the InfoRS that counters the timing issue of the greedy algorithm. Adding the information threshold in InfoRS avoids duplicate points in the memory, improving robustness against imbalanced data streams. Moreover, our proposed algorithms run efficiently, thanks to the proposed Bayesian model.

ACKNOWLEDGMENTS

We thank Yutian Chen, Razvan Pascanu and Yee Whye Teh for their constructive feedback.

ETHICS STATEMENT

This paper introduces a novel and efficient mechanism for selecting examples into an online memory. The proposed memory selection approach is applied to continual learning. An important benefit of this approach is that it allows a neural network to be trained from a stream of data, in which case it is not required to store all the data into a storage. The overall machine learning efficiency could be significantly improved in terms of both space storage and computation. However, potential risk comes with the fact that a small amount of information is selected into an online memory. Special care should be taken to ensure the user's privacy is not violated in certain situations.

REPRODUCIBILITY STATEMENT

Here we discuss our efforts to facilitate the reproducibility of the paper. Firstly, we present detailed pseudocodes for the proposed InfoRS (Alg 1) and InfoGS (Alg 2). We also present pseudocodes for the baselines used in the experiments, including reservoir sampling (Alg 3), weighted reservoir sampling (Alg 4), and class-balanced reservoir sampling (Alg 5). Moreover, we introduce the process to tune the hyper-parameters in the experimental section, and we present in Table 1 the detailed hyper-parameters we tuned for the involved approaches. In addition, we show detailed derivations and proofs in Sec E. Finally, besides the visualization figures, we present the precise means and standard errors of the experiments in Table 2.

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

# A  ALGORITHMS AND PSEUDOCODES

In this section we present more algorithms details and pseudocodes for the proposed InfoGS and the reservoir sampling.

## A.1  INFOGS

**Information improvement thresholding.** Let $(\mathbf{x}_{b^\star}, y_{b^\star}) \in \mathcal{M}$ be the memory point with the smallest MIC. We decide whether to replace it with the new point $(\mathbf{x}_\star, y_\star)$ by the information improvement thresholding. Specifically, let $\hat{\mu}_i, \hat{\sigma}_i$ be the running mean and standard deviation of the MIC for all historical observations, and let $\gamma_i$ be a hyper-parameter. The information improvement thresholding determines whether the new point improves the information by at least a certain amount,

$$\text{if:} \quad \text{MIC}_\eta((\mathbf{x}_\star, y_\star); \mathcal{M}) \geq \text{MIC}_\eta((\mathbf{x}_{b^\star}, y_{b^\star}); \mathcal{M}_{*,-b^\star}) + \hat{\mu}_i + \gamma_i * \hat{\sigma}_i. \tag{12}$$

**Learnability thresholding.** Outliers usually come with a large surprise. To further exclude outliers, we propose the learnability thresholding. Specifically, let $\hat{\mu}_l, \hat{\sigma}_l$ be the running mean and standard deviation of the learnability for all historical observations, and let $\gamma_l$ be a hyper-parameter. Then the learnability thresholding determines whether $s_{\text{learn}}((\mathbf{x}_\star, y_\star); \mathcal{M}) \geq \hat{\mu}_l + \gamma_l * \hat{\sigma}_l$. In this way, we deliberately take a certain percentage of data with low learnability out of consideration.

The overall algorithm, the Information-theoretic Greedy Selection (InfoGS), replaces the memory point $(\mathbf{x}_{b^\star}, y_{b^\star})$ with $(\mathbf{x}_\star, y_\star)$, if the new point passes both the information improvement thresholding and learnability thresholding. A pseudocode is shown in Alg 2.

---

**Algorithm 2** Information-theoretic Greedy Selection (InfoGS)

---

1: **Input:** Memory $\mathcal{M}$, the batch $\mathcal{B}^t$, the budget $M$, the predictor $f_{\boldsymbol{\theta}}$.
2: **Input:** Running mean and stddev for the information criterion: $\hat{\mu}_i, \hat{\sigma}_i$. The thresholding ratio $\gamma_i$.
3: **Input:** Running mean and stddev for the learnability: $\hat{\mu}_l, \hat{\sigma}_l$. The thresholding ratio $\gamma_l$.
4: Update $f_{\boldsymbol{\theta}}$ based on $\mathcal{M}$ and $(\mathbf{x}_\star, y_\star)$, with a network learning algorithm.
5: Compute $\mathbf{A}_{\mathcal{M}}^{-1}$ and $\mathbf{b}_{\mathcal{M}}$ using the features and targets in the memory.
6: **for** $_- = 1, ..., |\mathcal{B}^t|$ **do**
7:    **if** $|\mathcal{M}| < M$
8:       Update the buffer: $\mathcal{M} \longleftarrow \mathcal{M} \cup (\mathbf{x}_{b^\star}, y_{b^\star})$.
9:       Update $\mathbf{A}_{\mathcal{M}}^{-1}$ and $\mathbf{b}_{\mathcal{M}}$ based on the Sherman-Morrison formula.
10:      **continue**
11:    Get the batch index with the largest information criterion among those with large learnability:

$$b^\star = \operatorname*{arg\,max}_{b: s_{learn}((\mathbf{x}_b, y_b); \mathcal{M}) \geq \hat{\mu}_l + \hat{\sigma}_l * \gamma_l} \text{MIC}_\eta((\mathbf{x}_b, y_b); \mathcal{M}). \tag{13}$$

12:    Get the memory index with the smallest information criterion:

$$m^\star = \operatorname*{arg\,min}_m \text{MIC}_\eta((\mathbf{x}_m, y_m); \mathcal{M} \cup (\mathbf{x}_{b^\star}, y_{b^\star}) \setminus (\mathbf{x}_m, y_m)).$$

13:    **if** $\text{MIC}_\eta((\mathbf{x}_b, y_b); \mathcal{M}) \geq \text{MIC}_\eta((\mathbf{x}_{m^\star}, y_{m^\star}); \mathcal{M} \cup (\mathbf{x}_{b^\star}, y_{b^\star}) \setminus (\mathbf{x}_m, y_m)) + (\hat{\mu}_i + \hat{\sigma}_i * \gamma_i)$
14:       Update the buffer: $\mathcal{M} \longleftarrow \mathcal{M} \cup (\mathbf{x}_{b^\star}, y_{b^\star}) \setminus (\mathbf{x}_m, y_m)$.
15:       Update $\mathbf{A}_{\mathcal{M}}^{-1}$ and $\mathbf{b}_{\mathcal{M}}$ using $x_{b^\star}$ and $x_{m^\star}$ based on the Shermann-Morrison formula.
16:    **else**
17:       Update $\hat{\mu}_i, \hat{\sigma}_i, \hat{\mu}_l, \hat{\sigma}_l$ using $\mathcal{B}^t$.
18:       **return** Buffer $\mathcal{M}$ and predictor $f_{\boldsymbol{\theta}}$, the statistics $\hat{\mu}_i, \hat{\sigma}_i, \hat{\mu}_l, \hat{\sigma}_l$.
19: Update $\hat{\mu}_i, \hat{\sigma}_i, \hat{\mu}_l, \hat{\sigma}_l$ using $\mathcal{B}^t$.
20: **return** Buffer $\mathcal{M}$ and predictor $f_{\boldsymbol{\theta}}$, the statistics $\hat{\mu}_i, \hat{\sigma}_i, \hat{\mu}_l, \hat{\sigma}_l$.

---

## A.2 RESERVOIR SAMPLING

We present the pseudocodes for reservoir sampling (Vitter, 1985), the weighted reservoir sampling (Chao, 1982; Efraimidis & Spirakis, 2006), and the class-balanced reservoir sampling (Chrysakis & Moens, 2020). Worth mentioning, the weighted reservoir sampling becomes equivalent to reservoir sampling if the score function is a constant function. The class-balanced reservoir sampling becomes equivalent to reservoir sampling if only one class is observed.

---

**Algorithm 3** Reservoir Sampling (Vitter, 1985)

---

1: **Input:** Buffer $\mathcal{M}$, the budget $M$, the count $n$, the new data point $(\mathbf{x}_\star, y_\star)$.
2: **if** $|\mathcal{M}| < M$
3:    $\mathcal{M} \longleftarrow \mathcal{M} \cup (\mathbf{x}_\star, y_\star)$.
4: **else**
5:    Generate a random integer $i$ within $1, ..., n + 1$.
6:    **if** $i \leq M$
7:       $\mathcal{M} \longleftarrow \mathcal{M} \cup (\mathbf{x}_\star, y_\star) \backslash \mathcal{M}_i$.
8: **return** $\mathcal{M}, n + 1$.

---

---

**Algorithm 4** Weighted Reservoir Sampling (Chao, 1982; Efraimidis & Spirakis, 2006)

---

1: **Input:** Buffer $\mathcal{M}$, budget $M$, the new data point $(\mathbf{x}_\star, y_\star)$.
2: **Input:** The score function $w$ and the accumulative score $\bar{w}$.
3: Compute the score $w_\star = w(\mathbf{x}_\star, y_\star)$.
4: Update the accumulative score $\bar{w} = \bar{w} + w_\star$.
5: **if** $|\mathcal{M}| < M$
6:    $\mathcal{M} \longleftarrow \mathcal{M} \cup (\mathbf{x}_\star, y_\star)$.
7: **else**
8:    Compute normalized score $\hat{w}_\star = \min(\frac{w_\star}{\bar{w}}, \frac{1}{M})$.
9:    Generate a random integer $i$ within $1, ..., M + 1$, based on the probability,

$$[\overbrace{\hat{w}_\star, ..., \hat{w}_\star}^{M}, 1 - M * \hat{w}_\star].$$

10:    **if** $i \leq M$
11:       $\mathcal{M} \longleftarrow \mathcal{M} \cup (\mathbf{x}_\star, y_\star) \backslash \mathcal{M}_i$.
12: **return** $\mathcal{M}, \bar{w}$.

---

---

**Algorithm 5** Class-Balanced Reservoir Sampling (Chrysakis & Moens, 2020)

---

1: **Input:** Buffer $\mathcal{M}$, the budget $M$, the new data point $(\mathbf{x}_\star, y_\star)$.
2: **Input:** The counts $\mathbf{n} \in \mathbb{R}^K$ for all $K$ classes.
3: **if** $|\mathcal{M}| < M$
4:    $\mathcal{M} \longleftarrow \mathcal{M} \cup (\mathbf{x}_\star, y_\star)$.
5: **else**
6:    Get the class index $k$ with the largest count.
7:    **if** $\mathbf{n}_k > \mathbf{n}_{y_\star}$
8:       Generate a random integer $i$ within $1, ..., \mathbf{n}_k$.
9:       Replace the $i$-th class-$k$ point in the memory with $(\mathbf{x}_\star, y_\star)$.
10:       Update the counter: $\mathbf{n}_{y_\star} = \mathbf{n}_{y_\star} + 1$.
11:   **else**
12:       Let $\mathcal{M}_{y_\star}$ be the sub-buffer for class-$y_\star$.
13:       Update $\mathcal{M}_{y_\star}$ using reservoir sampling,

$$\mathcal{M}_{y_\star}, \mathbf{n}_{y_\star} \longleftarrow \textbf{ReservoirSampling}(\mathcal{M}_{y_\star}, |\mathcal{M}_{y_\star}|, \mathbf{n}_{y_\star}, (\mathbf{x}_\star, y_\star)).$$

14: **return** $\mathcal{M}, \mathbf{n}$.

---

## B  EXPERIMENTAL DETAILS

**Methods.** Our experiments involve Reserovir sampling (RS) (Vitter, 1985), weighted reservoir sampling with the output-space Hessian (WRS), class-balanced reservoir sampling (CBRS) (Chrysakis & Moens, 2020), greedy gradient sample selection (GSS) (Aljundi et al., 2019b), feature-space clustering (FSS-Clust) (Aljundi et al., 2019b), coreset (Borsos et al., 2020), InfoGS, and InfoRS.

**The toy Gaussian process regression.** We present here the experimental details of the toy GP regression in Figure 1. We consider a GP with the one-dimensional RBF kernel: $k(x, x') = \exp(-2(x - x')^2)$. We fix the variance of the observation noise as $0.04$. For the memory points, we generate the inputs using np.linspace(-1, 1, 10). We randomly sample noisy function values from the Gaussian process. For two new points, we fix them as $(0, 1)$ in the left and $(1.5, 1)$ in the right.

**Continual learning benchmarks.** To create data imbalance, we deliberately tie the task-ID that receives more epochs with the random seed, so that the imbalance appears for all tasks across the random runs. For example, in Split-CIFAR10, we make sure each of the five tasks is chosen twice within 10 random runs. We present the hyper-parameters and the configurations for the continual learning experiments in Table 1. We denote the batch size as *bs*, and the memory batch size as *mbs*. Specifically, we randomly sample *mbs* points from the memory buffer in each iteration to regularize the network. We denote the memory budget as $M$. For all the experiments, we adopt the stochastic gradient descent optimizer and we denote the learning rate as *lr*. Furthermore, $\alpha$ and $\beta$ are the regularization coefficients in the dark experience replay objective; $\eta$ is the coefficient of the learnability in the MIC; $\gamma_i$ is the coefficient of the standard deviation for the information improvement thresholding (InfoGS) or the information thresholding (InfoRS); $\gamma_l$ is the coefficient of the standard deviation for the learnability thresholding (InfoGS).

Table 1: Hyperparameters and Configurations of the Continual Learning Experiments.

| Permuted MNIST and Split MNIST | |
| --- | --- |
| Configuration | Network: FC-[100,100], base epoch $n_e = 1$, bs: 128, mbs: 128, M=100, sgd |
| Methods | Hyper-parameters |
| RS, WRS(var), CBRS | lr: $[0.03, 0.1, 0.3], \alpha : [0.3, 1.], \beta : [0.3, 1.]$ |
| GSS, FSS-Clust | lr: $[0.03, 0.1, 0.3], \alpha : [0.3, 1.], \beta : [0.3, 1.]$, Memory batch size for gss: 10 |
| Coreset | lr: $[0.001, 0.003, 0.01, 0.03], \alpha : [0.1, 0.3, 1.], \beta : [0.3]$, nr_slots : 10 |
| InfoGS | lr: $[0.03, 0.1, 0.3], \alpha : [0.3, 1.], \beta : [1.],$ $\eta : [0., 1., 3.], \gamma_i : [-0.3, 0, 0.3], \gamma_l : [0., 1.],$ noise sigma: $\sigma = 0.3$, jitter $c = \frac{\sigma^2}{\sigma_w^2} = 0.1$ |
| InfoRS | lr: $[0.03, 0.1, 0.3], \alpha : [0.3, 1.], \beta : [1.], \eta : [0., 1., 3.], \gamma_i : [-0.3, 0, 0.3]$ noise sigma: $\sigma = 0.3$, jitter $c = \frac{\sigma^2}{\sigma_w^2} = 0.1$ |
| Split CIFAR10 and Split MiniImageNet | |
| Configuration | Network: ResNet-18, bs: 32, mbs: 32, optimizer: sgd base epoch $n_e = 50$ (CIFAR10) and $n_e = 100$ (MiniImageNet) $M = 200$ (CIFAR10) and $M = 1000$ (MiniImageNet), data aug: [pad(4), random_crop, random_horizontal_flip] |
| Methods | Hyper-parameters |
| RS, WRS(var), CBRS | lr: $[0.01, 0.03], \alpha : [0.3, 1.], \beta : [1., 3.]$ |
| InfoGS | lr: $[0.01, 0.03], \alpha : [0.3, 1.], \beta : [1., 3.], \eta : [1.], \gamma_i : [-0.5, 0, 0.5], \gamma_l : [0.],$ noise sigma: $\sigma = 0.3$, jitter $c = \frac{\sigma^2}{\sigma_w^2} = 0.1$ |
| InfoRS | lr: $[0.01, 0.03], \alpha : [0.3, 1.], \beta : [1., 3.], \eta : [0., 1., 3.], \gamma_i : [-0.3, 0, 0.3]$ noise sigma: $\sigma = 0.3$, jitter $c = \frac{\sigma^2}{\sigma_w^2} = 0.1$ |

## C  INFORMATION-THEORETIC CRITERIA

In this section we present other information-theoretic criteria for online memory selection. Firstly, we have introduced the memorable information criterion composed of the learnability and the surprise,

$$\mathrm{MIC}_\eta((\mathbf{x}_\star, y_\star); \mathcal{M}) = \eta \log p(y'_\star | y_\star, \mathbf{y}_\mathcal{M}; \mathbf{X}_\mathcal{M}, \mathbf{x}_\star)|_{y'_\star = y_\star} - \log p(y_\star | \mathbf{y}_\mathcal{M}; \mathbf{X}_\mathcal{M}). \quad (14)$$

**Weighted Information Gain.** The standard information gain (Cover, 1999) is the KL divergence between the updated posterior and the memory posterior, which is further rewritten in Eq 7. Based on the rewritten form in Eq 7, we generalize the information gain as a weighted combination,

$$\mathrm{IG}_\eta((\mathbf{x}_\star, y_\star); \mathcal{M}) = \eta \mathbb{E}_{p(\mathbf{w}|y_\star, \mathbf{y}_\mathcal{M}; \mathbf{X}_\mathcal{M}, \mathbf{x}_\star)} \left[ \log p(y_\star | \mathbf{w}; \mathbf{x}_\star) \right] - \log p(y_\star | \mathbf{y}_\mathcal{M}; \mathbf{X}_\mathcal{M}), \quad (15)$$

where we include a coefficient $\eta$ to the first term similarly to MIC. We note that $\mathrm{IG}_1 = \mathrm{IG}$. Comparing $\mathrm{IG}_\eta$ and $\mathrm{MIC}_\eta$, we observe they share the same second surprise term, but differ in the first term. Nevertheless, observing that the first term in $\mathrm{IG}_\eta$ is the expected log probability of the new data point under the updated posterior $p(\mathbf{w}|y_\star, \mathbf{y}_\mathcal{M}; \mathbf{X}_\mathcal{M}, \mathbf{x}_\star)$, thus it quantifies the "learnability" as well but in a different form as in $\mathrm{MIC}_\eta$. Therefore, both $\mathrm{MIC}_\eta$ and $\mathrm{IG}_\eta$ are weighted combinations of "learnability" and "surprise".

Additionally, under the proposed Bayesian model in Subsec 2.3, the weighted IG can be computed in explicit forms as well,

$$\begin{aligned} \mathrm{IG}_\eta((\mathbf{x}_\star, y_\star); \mathcal{M}) =& \eta \log \mathcal{N}(y_\star | \mathbf{h}_\star^\top \mathbf{A}_{\mathcal{M}^+}^{-1} \mathbf{b}_{\mathcal{M}^+}, \sigma^2) - \frac{\eta}{2} \mathbf{h}_\star^\top \mathbf{A}_{\mathcal{M}^+}^{-1} \mathbf{h}_\star \\ & - \log \mathcal{N}(y_\star | \mathbf{h}_\star^\top \mathbf{A}_\mathcal{M}^{-1} \mathbf{b}_\mathcal{M}, \sigma^2 \mathbf{h}_\star^\top \mathbf{A}_\mathcal{M}^{-1} \mathbf{h}_\star + \sigma^2), \end{aligned} \quad (16)$$

where $\mathbf{h}_\star$ is the feature of $\mathbf{x}_\star$.

**Entropy Reduction.** The entropy reflects the amount of information carried in a distribution. Thus another information-theoretic criterion would be how much the entropy can be reduced by adding the new point into the buffer. We propose the entropy reduction criterion as the following,

$$\mathrm{ER}((\mathbf{x}_\star, y_\star); \mathcal{M}) := \mathbb{H}[p(\mathbf{w}|\mathcal{M})] - \mathbb{H}[p(\mathbf{w}|\mathcal{M}, (\mathbf{x}, y))], \quad (17)$$

where $\mathbb{H}$ represents the entropy of the distribution. The weight posterior is a Gaussian distribution under the Bayesian model in Subsec 2.3, thus the entropy can be computed explicitly as well,

$$\begin{aligned} \mathrm{ER}((\mathbf{x}_\star, y_\star); \mathcal{M}) &= \frac{1}{2} \log|\mathbf{A}_\mathcal{M}^{-1}| - \frac{1}{2} \log|\mathbf{A}_{\mathcal{M}^+}^{-1}| = \frac{1}{2} \log|(\mathbf{A}_\mathcal{M} + \mathbf{x}_\star \mathbf{x}_\star^\top) \mathbf{A}_\mathcal{M}^{-1}| \\ &= \frac{1}{2} \log|\mathbf{I} + \mathbf{x}_\star \mathbf{x}_\star^\top \mathbf{A}_\mathcal{M}^{-1}| = \frac{1}{2} \log(1 + \mathbf{x}_\star^\top \mathbf{A}_\mathcal{M}^{-1} \mathbf{x}_\star). \end{aligned} \quad (18)$$

We note that this entropy reduction is independent of the target $y_\star$ of the new point.

**Comparing the information-theoretic criteria.** Intuitively, we propose the "surprise" to detect unexpected points for gathering the most information in the memory, and we propose the "learnability" to encourage the consensus between the new point and the memory. As demonstrated by $\mathrm{MIC}_\eta$ and $\mathrm{IG}_\eta$, the "learnability" can be quantified by different explicit expressions. Similarly, the "surprise" criterion might have different expressions as well. This is also illustrated by the prediction gain (Graves et al., 2017), where the losses before and after one step of gradient updates can be understood as the "surprise" and "learnability", respectively. Therefore, we do not think that $\mathrm{MIC}_\eta$ strictly outperforms $\mathrm{IG}_\eta$, since both combine the "surprise" and "learnability". In this paper we pick the $\mathrm{MIC}_\eta$ since its learnability and surprise expressions are in a more comparable form, given that both are log predictive densities.

The entropy reduction criterion does not depend on the target $y_\star$ under the Gaussian weight posterior. Unlike the "surprise" criterion, ER encourages the diversity of inputs. Therefore it can behave more like an unsuperised information criterion where output/label information is not captured. For example, if the underlying distribution is composed of two input modes (where each mode has different target labels), while one mode is larger in the input space but the other mode is smaller. Then ER might focus more on the larger mode but neglects the smaller mode, yet this problem could be fixed by considering the target observations.

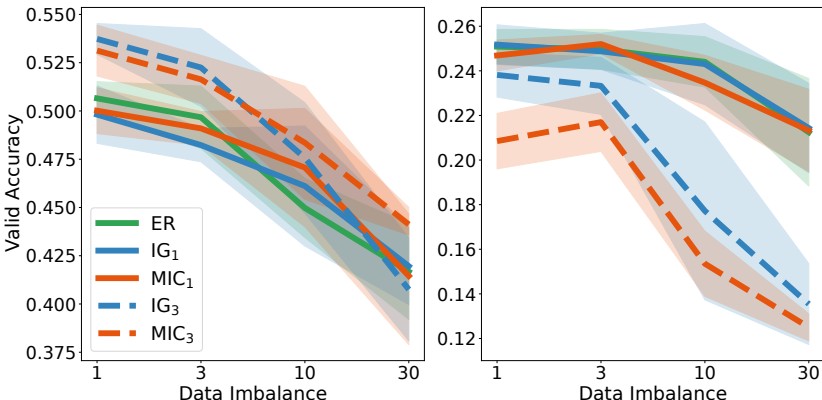

Figure 7: The performances of InfoRS with varying criteria for Split CIFAR10 (*left*) and Split MiniImageNet (*right*). We compare $ER, MIC_1, MIC_3, IG_1, IG_3$.

We further conduct Split CIFAR10 and Split MiniImageNet experiments to compare $ER, IG_1, IG_3, MIC_1, MIC_3$. The results are shown in Figure 7. We observe $IG_\eta$ and $MIC_\eta$ perform similarly in this experiment. Furthermore, we observe that using $\eta = 3$ improves over $\eta = 1$ for the Split CIFAR10 experiment, while using $\eta = 1$ improves over $\eta = 3$ for the Split MiniImageNet experiment. This indicates the importance to balance the surprise and learnability in the information-theoretic criteria. We also observe that the entropy reduction achieves competitive results as well, although it does not rely on the targets of new points.

## D    MORE EXPERIMENTS

In this section we present more experimental results. Firstly, in Table 2 we present the test performances of the continual learning benchmarks, accompanying Figure 4.

**More baselines.** We compare the proposed approaches with the gradient sample selection (GSS) (Aljundi et al., 2019b) and class-balanced reservoir sampling (CBRS) (Chaudhry et al., 2019a) as well. We choose the GSS-Greedy algorithm in Aljundi et al. (2019b), which proposes a greedy algorithm to minimize gradient similarities in the memory. However, since evaluating the per-example gradients in each iteration causes large computational costs, we only managed to run it for Permuted MNIST and Split MNIST. As argued in the paper, RS is vulnerable to imbalanced data streams. Specifically for classifications, RS tends to select the classes with more appearances and leads to an imbalanced buffer. Class-balanced reservoir sampling (Chaudhry et al., 2019a) proposes to force a balanced buffer across classes by conducting reservoir sampling separately for each class. Consequently, when $K$ classes are observed, each class will occupy $M/K$ points in the memory. A pseudocode for CBRS is shown in Alg 5. However, CBRS is only applicable to fix the class-imbalance problem for classifications. In comparison, InfoRS is generally applicable to all forms of data imbalance in the problems.

We include GSS and CBRS in Figure 8. We observe that GSS performs worse than other approaches. For the CBRS, we first observe that it sees a performance degradation similar to RS over Permuted MNIST, since CBRS can only fix the class-imbalance problem, but the classes in Permuted MNIST are always balanced. For the other three benchmarks where class-imbalance appears, CBRS demonstrates the best robustness compared to all approaches. However, as observed in Split CIFAR10 and Split MiniImageNet, CBRS underperforms RS and InfoRS when the data is balanced. We argue that the timing of the memory updates influences CBRS here. Because CBRS attempts to maintain a class-balanced buffer, it will immediately include new points into the buffer when these points are from previously unseen classes. In consequence, CBRS tends to update the buffer at the beginning of each task for the three Split benchmarks. Then the historical points will be removed earlier. In particular, for the dark experience replay, the stored logits at the beginning of each task will be harmful to network learning as well, which also affects the performance of CBRS.

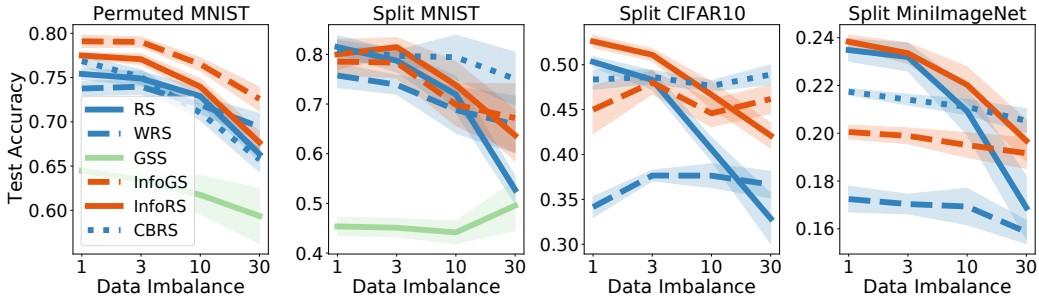

Figure 8: Test performances against data imbalances over continual learning benchmarks. We compare RS, weighted RS (WRS) using the output-space Hessian, InfoGS, InfoRS, GSS, and CBRS. For each experiment, we plot the mean and the 95% confidence interval across 10 random seeds.

We also include the feature-space clustering method (FSS-Clust) (Aljundi et al., 2019b) for Permuted MNIST and Split MNIST. The results are shown in Table 2. We observe that FSS-Clust also underperforms RS and InfoRS.

**Ablation studies for InfoGS and InfoRS.** InfoRS is motivated by trying to resolve the timing issue of InfoGS. However, besides the difference of *"when to update the memory"* between InfoRS and InfoGS, *"how to update the memory"* is still different since InfoRS removes a random memory point while InfoGS removes the least informative memory point. Therefore, we further present an ablation algorithm to separate the effect of *"when to update the memory"* and *"how to update the memory"*. In InfoGS, the new point is added to the memory as long as it passes the *information improvement thresholding* and the *learnability thresholding*, thus InfoGS tends to update the memory urgently. To bridge the gap between InfoGS and InfoRS, we propose to decide *"when to update the memory"* by reservoir sampling as well. Specifically, if the new point passes both thresholding in InfoGS, we decide whether to include it in the memory by reservoir sampling. If the point also passes the reservoir sampling, we put the new point into the memory and remove the least informative existing point. The resulting algorithm is represented InfoGS-RS.

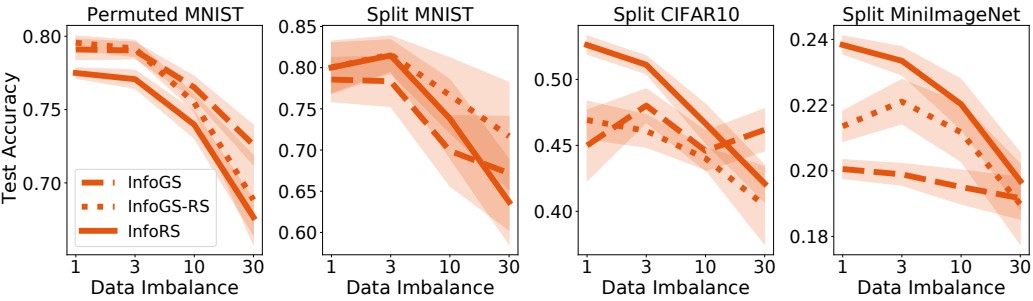

Figure 9: Test performances against data imbalances over continual learning benchmarks. We compare InfoGS, InfoRS, and the InfoGS with reservoir sampling (InfoGS-RS).

We compare InfoGS, InfoRS and InfoGS-RS in Figure 9. From the figure we observe that although InfoGS-RS incorporates reservoir sampling to remedy the timing issue, it still underperforms InfoRS when the tasks are balanced. We hypothesize that feature learning might be accounted for the difference. For example, due to the effect of regularizations, the features for the memory points tend to be stable in the learning process, while the features for new points will adapt to learn the underlying mapping. Due to the evolving features, new points might have a larger chance to be surprising and existing memories could be deleted as a result.

**Forward transfer and backward transfer.** We also present the forward transfer and backward transfer for the continual learning benchmarks in Table 3 and Table 4, respectively. The forward and backward transfer is introduced in Lopez-Paz & Ranzato (2017). Specifically, let $R_{i,j}$ be the

Table 2: The means and standard errors of the continual learning benchmarks.

| Method | imbalance=1 | imbalance=3 | imbalance=10 | imbalance=30 |
|---|---|---|---|---|
| | Permuted MNIST (M=100) | | | |
| RS (Buzzega et al., 2020) | $75.41 \pm 0.32$ | $74.90 \pm 0.42$ | $72.90 \pm 0.47$ | $66.35 \pm 0.41$ |
| WRS | $73.76 \pm 0.41$ | $73.97 \pm 0.43$ | $71.92 \pm 0.50$ | $69.44 \pm 0.77$ |
| CBRS (Chrysakis & Moens, 2020) | $76.86 \pm 0.31$ | $75.07 \pm 0.42$ | $71.02 \pm 0.44$ | $65.73 \pm 0.72$ |
| GSS (Aljundi et al., 2019b) | $64.46 \pm 0.48$ | $63.45 \pm 0.64$ | $61.74 \pm 1.11$ | $59.36 \pm 1.59$ |
| FSS-Clust (Aljundi et al., 2019b) | $70.53 \pm 0.33$ | $70.12 \pm 0.57$ | $68.35 \pm 0.82$ | $65.82 \pm 0.90$ |
| InfoGS | $\mathbf{79.10 \pm 0.36}$ | $\mathbf{79.03 \pm 0.29}$ | $\mathbf{76.51 \pm 0.39}$ | $\mathbf{72.59 \pm 0.69}$ |
| InfoRS | $77.50 \pm 0.19$ | $77.07 \pm 0.32$ | $73.97 \pm 0.48$ | $67.67 \pm 0.62$ |
| | Split MNIST (M=100) | | | |
| RS (Buzzega et al., 2020) | $\mathbf{81.49 \pm 0.72}$ | $78.74 \pm 1.17$ | $71.94 \pm 1.87$ | $52.76 \pm 1.41$ |
| WRS | $75.75 \pm 1.24$ | $73.90 \pm 0.97$ | $68.82 \pm 2.39$ | $65.87 \pm 2.92$ |
| CBRS(Chrysakis & Moens, 2020) | $80.96 \pm 1.43$ | $79.71 \pm 0.55$ | $\mathbf{79.43 \pm 2.28}$ | $\mathbf{75.08 \pm 2.73}$ |
| GSS (Aljundi et al., 2019b) | $45.36 \pm 0.95$ | $45.10 \pm 0.96$ | $44.16 \pm 1.15$ | $49.54 \pm 2.57$ |
| FSS-Clust (Aljundi et al., 2019b) | $52.85 \pm 1.13$ | $54.59 \pm 1.69$ | $54.36 \pm 2.11$ | $55.33 \pm 2.89$ |
| Coreset (Borsos et al., 2020) | $69.77 \pm 0.75$ | $63.47 \pm 2.21$ | $50.30 \pm 2.97$ | $42.05 \pm 2.19$ |
| InfoGS | $78.55 \pm 1.37$ | $78.35 \pm 1.56$ | $69.90 \pm 2.18$ | $67.19 \pm 3.52$ |
| InfoRS | $80.02 \pm 1.53$ | $\mathbf{81.48 \pm 0.98}$ | $73.81 \pm 2.43$ | $63.70 \pm 2.63$ |
| | Split CIFAR10 (M=200) | | | |
| RS(Buzzega et al., 2020) | $50.32 \pm 0.30$ | $48.31 \pm 0.54$ | $40.59 \pm 0.67$ | $32.88 \pm 1.43$ |
| WRS | $34.14 \pm 0.59$ | $37.66 \pm 0.30$ | $37.64 \pm 0.68$ | $36.65 \pm 0.76$ |
| CBRS(Chrysakis & Moens, 2020) | $48.35 \pm 0.55$ | $48.64 \pm 0.39$ | $\mathbf{47.65 \pm 0.31}$ | $\mathbf{48.90 \pm 0.56}$ |
| InfoGS | $44.99 \pm 1.37$ | $48.02 \pm 0.65$ | $44.59 \pm 0.77$ | $46.18 \pm 0.82$ |
| InfoRS | $\mathbf{52.61 \pm 0.36}$ | $\mathbf{51.11 \pm 0.35}$ | $46.66 \pm 0.76$ | $42.10 \pm 0.71$ |
| | Split MiniImageNet (M=1000) | | | |
| RS(Buzzega et al., 2020) | $23.48 \pm 0.24$ | $23.19 \pm 0.30$ | $20.93 \pm 0.51$ | $16.89 \pm 0.67$ |
| WRS | $17.24 \pm 0.27$ | $17.04 \pm 0.21$ | $16.93 \pm 0.39$ | $15.87 \pm 0.25$ |
| CBRS(Chrysakis & Moens, 2020) | $21.74 \pm 0.07$ | $21.41 \pm 0.11$ | $21.10 \pm 0.16$ | $\mathbf{20.52 \pm 0.26}$ |
| InfoGS | $20.05 \pm 0.15$ | $19.89 \pm 0.17$ | $19.51 \pm 0.26$ | $19.16 \pm 0.33$ |
| InfoRS | $\mathbf{23.83 \pm 0.14}$ | $\mathbf{23.35 \pm 0.22}$ | $\mathbf{22.03 \pm 0.39}$ | $19.69 \pm 0.44$ |

performance of task $j$ after training task $i$, and let $b_i$ be the performance of task $i$ at initialization, then they are defined in the following expression,

$$\text{backward transfer} = \frac{1}{T-1} \sum_{i=1}^{T-1} (R_{T,i} - R_{i,i})$$

$$\text{forward transfer} = \frac{1}{T-1} \sum_{i=1}^{T-1} (R_{i,i+1} - b_{i+1}).$$

Intuitively, backward transfer measures how training new tasks can improve previous performances. Forward transfer measures how much performance improvement over a random network can be achieved by training previous tasks. New classes appear in news tasks in Split benchmarks, making the forward transfer is almost impossible. In contrast, it is possible for the network to explore the relationship between data points in the Permuted MNIST to achieve forward transfer. Therefore, we only report forward transfer for Permuted MNIST and Split MNIST for comparison, while we report backward transfer for all benchmarks.

Table 3: The means and standard errors of forward transfers in continual learning benchmarks.

| Method | imbalance=1 | imbalance=3 | imbalance=10 | imbalance=30 |
|---|---|---|---|---|
| | Permuted MNIST (M=100) | | | |
| RS(Buzzega et al., 2020) | $0.88 \pm 0.27$ | $0.73 \pm 0.24$ | $0.74 \pm 0.24$ | $0.75 \pm 0.27$ |
| WRS | $0.86 \pm 0.28$ | $0.65 \pm 0.22$ | $0.84 \pm 0.25$ | $0.80 \pm 0.25$ |
| CBRS(Chrysakis & Moens, 2020) | $0.90 \pm 0.22$ | $0.84 \pm 0.30$ | $0.72 \pm 0.33$ | $0.81 \pm 0.22$ |
| GSS (Aljundi et al., 2019b) | $0.74 \pm 0.26$ | $0.64 \pm 0.29$ | $0.77 \pm 0.25$ | $0.55 \pm 0.25$ |
| FSS-Clust (Aljundi et al., 2019b) | $1.91 \pm 0.23$ | $1.89 \pm 0.28$ | $1.74 \pm 0.28$ | $1.74 \pm 0.34$ |
| InfoGS | $0.75 \pm 0.26$ | $0.67 \pm 0.32$ | $0.78 \pm 0.29$ | $0.58 \pm 0.28$ |
| InfoRS | $0.64 \pm 0.30$ | $0.92 \pm 0.25$ | $0.82 \pm 0.25$ | $0.58 \pm 0.23$ |
| | Split MNIST (M=100) | | | |
| RS(Buzzega et al., 2020) | $-9.68 \pm 0.89$ | $-9.68 \pm 0.89$ | $-9.68 \pm 0.89$ | $-9.68 \pm 0.89$ |
| WRS | $-9.68 \pm 0.89$ | $-9.68 \pm 0.89$ | $-9.68 \pm 0.89$ | $-9.68 \pm 0.89$ |
| CBRS(Chrysakis & Moens, 2020) | $-9.68 \pm 0.89$ | $-9.68 \pm 0.89$ | $-9.68 \pm 0.89$ | $-9.68 \pm 0.89$ |
| GSS (Aljundi et al., 2019b) | $-9.68 \pm 0.89$ | $-9.68 \pm 0.89$ | $-9.68 \pm 0.89$ | $-9.68 \pm 0.89$ |
| FSS-Clust (Aljundi et al., 2019b) | $-10.09 \pm 1.58$ | $-10.09 \pm 1.58$ | $-10.09 \pm 1.58$ | $-10.09 \pm 1.58$ |
| Coreset (Borsos et al., 2020) | $-10.09 \pm 1.58$ | $-10.09 \pm 1.58$ | $-10.09 \pm 1.58$ | $-10.09 \pm 1.58$ |
| InfoGS | $-9.68 \pm 0.89$ | $-9.68 \pm 0.89$ | $-9.68 \pm 0.89$ | $-9.68 \pm 0.89$ |
| InfoRS | $-9.68 \pm 0.89$ | $-9.68 \pm 0.89$ | $-9.68 \pm 0.89$ | $-9.68 \pm 0.89$ |

Table 4: The means and standard errors of backward transfers in the continual learning benchmarks.

| Method | imbalance=1 | imbalance=3 | imbalance=10 | imbalance=30 |
|---|---|---|---|---|
| | Permuted MNIST (M=100) | | | |
| RS(Buzzega et al., 2020) | $-19.27 \pm 0.26$ | $-21.30 \pm 0.31$ | $-23.34 \pm 0.30$ | $-31.48 \pm 0.43$ |
| WRS | $-21.69 \pm 0.30$ | $-21.98 \pm 0.58$ | $-24.60 \pm 0.72$ | $-28.07 \pm 0.76$ |
| CBRS(Chrysakis & Moens, 2020) | $-19.98 \pm 0.43$ | $-21.30 \pm 0.39$ | $-24.72 \pm 0.31$ | $-30.43 \pm 0.65$ |
| GSS (Aljundi et al., 2019b) | $-32.58 \pm 0.57$ | $-33.53 \pm 0.74$ | $-34.55 \pm 0.96$ | $-36.23 \pm 1.22$ |
| FSS-Clust (Aljundi et al., 2019b) | $-24.72 \pm 0.34$ | $-25.42 \pm 0.59$ | $-27.38 \pm 0.82$ | $-30.14 \pm 0.90$ |
| InfoGS | $\mathbf{-15.68 \pm 0.30}$ | $\mathbf{-17.20 \pm 0.26}$ | $\mathbf{-19.58 \pm 0.28}$ | $\mathbf{-22.74 \pm 0.62}$ |
| InfoRS | $-18.16 \pm 0.22$ | $-20.20 \pm 0.52$ | $-22.44 \pm 0.51$ | $-28.95 \pm 0.57$ |
| | Split MNIST (M=100) | | | |
| RS(Buzzega et al., 2020) | $-21.90 \pm 1.53$ | $-25.58 \pm 1.36$ | $-36.75 \pm 1.88$ | $-53.51 \pm 1.32$ |
| WRS | $-24.33 \pm 1.99$ | $-26.03 \pm 1.36$ | $-37.78 \pm 3.19$ | $-39.65 \pm 4.42$ |
| CBRS(Chrysakis & Moens, 2020) | $-22.62 \pm 1.29$ | $\mathbf{-23.75 \pm 1.02}$ | $\mathbf{-25.33 \pm 2.67}$ | $\mathbf{-25.88 \pm 3.62}$ |
| GSS (Aljundi et al., 2019b) | $-65.27 \pm 1.25$ | $-65.58 \pm 1.43$ | $-65.59 \pm 1.33$ | $-62.55 \pm 2.17$ |
| FSS-Clust (Aljundi et al., 2019b) | $-56.49 \pm 1.45$ | $-54.52 \pm 2.14$ | $-53.10 \pm 3.27$ | $-54.19 \pm 3.67$ |
| Coreset (Borsos et al., 2020) | $-18.72 \pm 0.79$ | $-34.04 \pm 2.88$ | $-50.00 \pm 4.29$ | $-59.13 \pm 3.74$ |
| InfoGS | $-21.62 \pm 1.93$ | $-28.59 \pm 1.99$ | $-33.11 \pm 3.25$ | $-34.81 \pm 5.12$ |
| InfoRS | $\mathbf{-18.52 \pm 1.06}$ | $-23.81 \pm 1.83$ | $-31.74 \pm 3.83$ | $-43.37 \pm 2.71$ |
| | Split CIFAR10 (M=200) | | | |
| RS(Buzzega et al., 2020) | $-44.09 \pm 0.75$ | $-50.74 \pm 1.53$ | $-53.00 \pm 2.26$ | $-52.66 \pm 4.51$ |
| WRS | $-61.35 \pm 1.03$ | $-58.45 \pm 0.96$ | $-55.91 \pm 0.95$ | $-56.33 \pm 1.51$ |
| CBRS(Chrysakis & Moens, 2020) | $-53.92 \pm 0.50$ | $-55.62 \pm 0.54$ | $-53.56 \pm 0.49$ | $-55.32 \pm 0.83$ |
| InfoGS | $-51.92 \pm 2.42$ | $\mathbf{-43.14 \pm 1.26}$ | $\mathbf{-42.80 \pm 1.19}$ | $\mathbf{-49.30 \pm 1.78}$ |
| InfoRS | $\mathbf{-43.46 \pm 0.66}$ | $-46.72 \pm 1.06$ | $-47.02 \pm 1.75$ | $-51.63 \pm 2.62$ |
| | Split MiniImageNet (M=1000) | | | |
| RS(Buzzega et al., 2020) | $\mathbf{-52.52 \pm 0.27}$ | $\mathbf{-52.68 \pm 0.59}$ | $\mathbf{-53.62 \pm 1.58}$ | $\mathbf{-52.67 \pm 3.09}$ |
| WRS | $-61.11 \pm 0.37$ | $-60.59 \pm 0.47$ | $-59.63 \pm 0.44$ | $-61.28 \pm 0.31$ |
| CBRS(Chrysakis & Moens, 2020) | $-58.33 \pm 0.15$ | $-57.91 \pm 0.17$ | $-58.91 \pm 0.29$ | $-59.34 \pm 0.23$ |
| InfoGS | $-58.80 \pm 0.13$ | $-59.14 \pm 0.32$ | $-59.87 \pm 0.34$ | $-60.91 \pm 0.31$ |
| InfoRS | $-53.04 \pm 0.23$ | $-53.41 \pm 0.62$ | $-53.89 \pm 1.27$ | $-57.28 \pm 1.06$ |

**Ablation studies for $\eta$ and $\gamma_i$.** The learnability ratio $\eta$ controls the weighting between surprise and learnability. We present an ablation study to investigate the impact of $\eta$ in InfoRS. In addition, the information thresholding parameter $\gamma_i$ determines the degree of InfoRS from acting purely randomly to greedily. We also present an ablation study to investigate its impact. The results are shown in Figure 10.

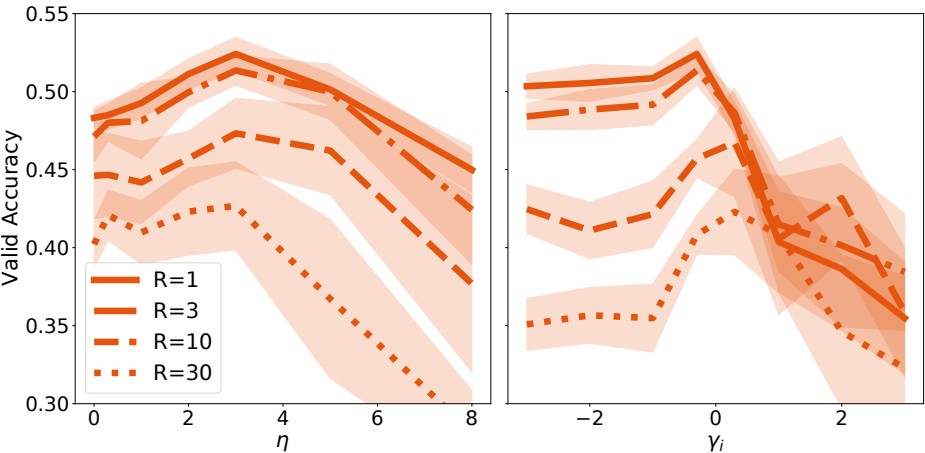

Figure 10: Validation performances against the learnability ratio ($\eta$) and the information thresholding ratio ($\gamma_i$), over Split CIFAR10. We compare InfoRS under various data imbalances ($R$) and plot the means and 95% confidence intervals. Firstly, for all data imbalance, the best $\eta$ is achieved at around 3 and the best $\gamma_i$ at around 0. Thus both hyperparameters are not sensitive to data imbalance. Secondly, for the learnability ratio, setting $\eta = 0$ and setting $\eta$ to be large both lead to degraded performances, which indicates the importance of properly balancing surprise and learnability in online memory selection. Thirdly, for the information thresholding ratio, setting it to be small makes InfoRS act similarly to RS while setting it to be large makes InfoRS more greedy. Therefore, we observe that when the data is balanced ($R = 1, R = 3$), setting $\gamma_i < 0$ leads to similar optimal performances. In contrast, when the data is imbalanced ($R = 10, R = 30$), choosing an intermediate $\gamma_i$ can properly balance the stochasticity and the greediness.

**Evolution of test accuracies along with training.** In Figure 11 we show how the test accuracies evolve along with seeing more tasks. Specifically, for each dataset, we visualize the evolutions in RS and InfoRS over a single random run, where the first task is trained for 30 times epochs than the others.

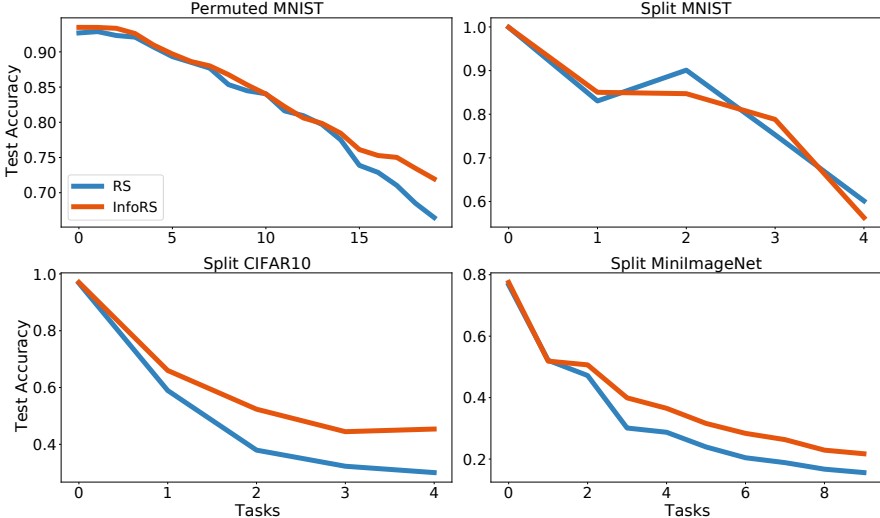

Figure 11: The evolution of test accuracies. The first task is trained for 30 times longer than others.

**The TSNE plot with a smaller memory.** In Figure 2, we demonstrated that incorporating learnability is important to filter out outliers, using the TSNE plot of 200 selected memories. Here we present another comparison to investigate the scenario with a small memory. Specifically, we set the memory buffer as 25. The results are shown in Figure 12.

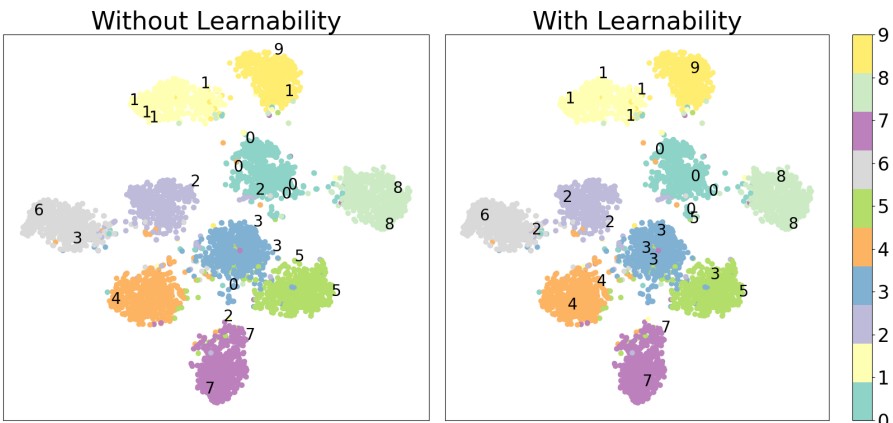

Figure 12: TSNE visualizations of the training data (colored dots, where the color represents the label of the data point) and 25 memory points (black digits, where the digit represents the label of the point) for InfoGS without and with learnability, respectively. With a small memory, we observe that InfoGS without learnability performs similarly to InfoGS with learnability. Both select a representative memory buffer that reflects the training distribution. Because the agent needs to select 25 memories out of data points from 10 classes, the surprise will dominate the selection process, and the outlier issue is not protruding. Therefore, incorporating learnability is not critical to the selection when the memory buffer is very small.

**Impact of $\alpha$ and $\beta$.** The logit regularization coefficient $\alpha$ and the label regularization coefficient $\beta$ in DER++ control the regularization degree of the replay buffer. We further present an experiment investigate their impact separately, in Figure 13.

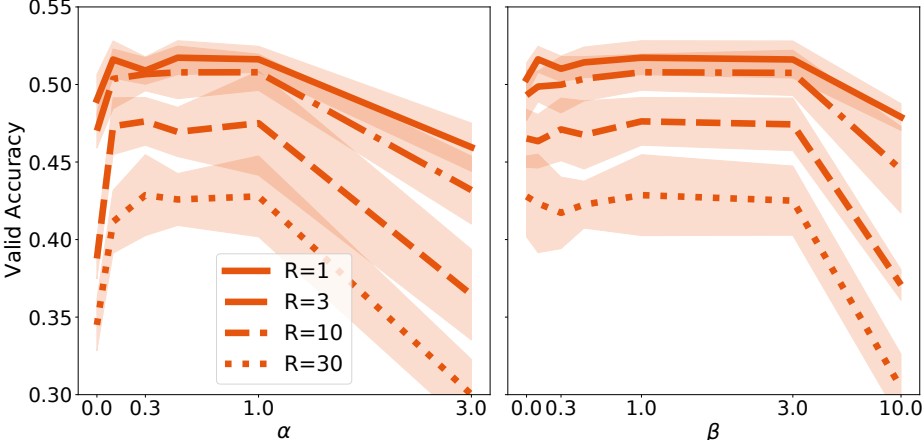

Figure 13: Validation performances against the logit regularization coefficient ($\alpha$) and the label regularization coefficient ($\beta$) in DER++, over Split CIFAR10. We compare InfoRS under various data imbalances ($R$) and plot the means and 95% confidence intervals. We observe that the performance is not sensitive to $\alpha$ and $\beta$.

# E  PROOFS

In this section we provide necessary proofs.

*Information Gain.* We rewrite the information gain as the following,

$$
\begin{aligned}
\mathrm{IG}((\mathbf{x}_\star, y_\star); \mathcal{M}) &= \mathrm{KL}\left(p(\mathbf{w}|y_\star, \mathbf{y}_\mathcal{M}; \mathbf{X}_\mathcal{M}, \mathbf{x}_\star) \| p(\mathbf{w}|\mathbf{y}_\mathcal{M}; \mathbf{X}_\mathcal{M})\right) \\
&= \mathbb{E}_{p(\mathbf{w}|y_\star, \mathbf{y}_\mathcal{M}; \mathbf{X}_\mathcal{M}, \mathbf{x}_\star)}\left[\log \frac{p(\mathbf{w}|y_\star, \mathbf{y}_\mathcal{M}; \mathbf{X}_\mathcal{M}, \mathbf{x}_\star)}{p(\mathbf{w}|\mathbf{y}_\mathcal{M}; \mathbf{X}_\mathcal{M})}\right] \\
&= \mathbb{E}_{p(\mathbf{w}|y_\star, \mathbf{y}_\mathcal{M}; \mathbf{X}_\mathcal{M}, \mathbf{x}_\star)}\left[\log \frac{p(\mathbf{w}, y_\star, \mathbf{y}_\mathcal{M}; \mathbf{X}_\mathcal{M}, \mathbf{x}_\star)}{p(\mathbf{w}, \mathbf{y}_\mathcal{M}; \mathbf{X}_\mathcal{M}) p(y_\star|\mathbf{y}_\mathcal{M}; \mathbf{X}_\mathcal{M}, \mathbf{x}_\star)}\right] \\
&= \mathbb{E}_{p(\mathbf{w}|y_\star, \mathbf{y}_\mathcal{M}; \mathbf{X}_\mathcal{M}, \mathbf{x}_\star)}[\log p(y_\star|\mathbf{w}, \mathbf{y}_\mathcal{M}) - \log p(y_\star|\mathbf{y}_\mathcal{M}; \mathbf{X}_\mathcal{M})] \\
&= \mathbb{E}_{p(\mathbf{w}|y_\star, \mathbf{y}_\mathcal{M}; \mathbf{X}_\mathcal{M}, \mathbf{x}_\star)}[\log p(y_\star|\mathbf{w}; \mathbf{x}_\star)] - \log p(y_\star|\mathbf{y}_\mathcal{M}; \mathbf{X}_\mathcal{M}). \quad (19)
\end{aligned}
$$

$\square$

*Proof.* By applying Jensen's inequality we have

$$
\begin{aligned}
\log p(y'_\star|y_\star, \mathbf{y}_\mathcal{M}; \mathbf{X}_\mathcal{M}, \mathbf{x}_\star)|_{y'_\star = y_\star} &= \log \mathbb{E}_{p(\mathbf{w}|y_\star, \mathbf{y}_\mathcal{M}; \mathbf{X}_\mathcal{M}, \mathbf{x}_\star)}[p(y_\star|\mathbf{w}; \mathbf{x}_\star)] \\
&\geq \mathbb{E}_{p(\mathbf{w}|y_\star, \mathbf{y}_\mathcal{M}; \mathbf{X}_\mathcal{M}, \mathbf{x}_\star)}[\log p(y_\star|\mathbf{w}; \mathbf{x}_\star)] \quad (20)
\end{aligned}
$$

$\square$

*Information Criterion for Bayesian Linear Regression.* Given a new point $(\mathbf{x}_\star, y_\star)$, let $\mathbf{h}_\star$ be the corresponding feature, then the criterion is represented as,

$$
s((\mathbf{h}_\star, y_\star); \mathcal{M}) = \eta \log p(y'_\star|y_\star, \mathbf{y}_\mathcal{M}; \mathbf{X}_\mathcal{M}, \mathbf{x}_\star)|_{y'_\star = y_\star} - \log p(y_\star|\mathbf{y}_\mathcal{M}; \mathbf{X}_\mathcal{M}). \quad (21)
$$

Given the memory $\mathcal{M}$, the posterior distribution of weights can be computed as,

$$
p(\mathbf{w}|\mathbf{y}_\mathcal{M}; \mathbf{H}_\mathcal{M}) = \mathcal{N}(\mathbf{A}_\mathcal{M}^{-1} \mathbf{b}_\mathcal{M}, \sigma^2 \mathbf{A}_\mathcal{M}^{-1}). \quad (22)
$$

where the matrix $\mathbf{A}_\mathcal{M}^{-1} = \left(\mathbf{H}_\mathcal{M}^\top \mathbf{H}_\mathcal{M} + c\mathbf{I}_d\right)^{-1}$ for $c = \frac{\sigma^2}{\sigma_w^2}$ and $\mathbf{b}_\mathcal{M} = \mathbf{H}_\mathcal{M}^\top \mathbf{y}_\mathcal{M}$. Similarly let $\mathcal{M}^+ = \mathcal{M} \cup (\mathbf{x}_\star, y_\star)$, $\mathbf{A}_{\mathcal{M}^+} = \mathbf{A}_\mathcal{M} + \mathbf{h}_\star \mathbf{h}_\star^\top$ and $\mathbf{b}_{\mathcal{M}^+} = \mathbf{b}_\mathcal{M} + \mathbf{h}_\star y_\star$, then weight posterior given the memory $\mathcal{M}^+$ can be computed as,

$$
p(\mathbf{w}|\mathbf{y}_{\mathcal{M}^+}; \mathbf{H}_{\mathcal{M}^+}) = \mathcal{N}(\mathbf{A}_{\mathcal{M}^+}^{-1} \mathbf{b}_{\mathcal{M}^+}, \sigma^2 \mathbf{A}_{\mathcal{M}^+}^{-1}). \quad (23)
$$

Thus the predictive distributions can be represented as,

$$
y'_\star|\mathbf{y}_{\mathcal{M}^+} \sim \mathcal{N}(\mathbf{h}_\star^\top \mathbf{A}_{\mathcal{M}^+}^{-1} \mathbf{b}_{\mathcal{M}^+}, \sigma^2 \mathbf{h}_\star^\top \mathbf{A}_{\mathcal{M}^+}^{-1} \mathbf{h}_\star + \sigma^2), \quad (24)
$$

$$
y_\star|\mathbf{y}_\mathcal{M} \sim \mathcal{N}(y_\star|\mathbf{h}_\star^\top \mathbf{A}_\mathcal{M}^{-1} \mathbf{b}_\mathcal{M}, \sigma^2 \mathbf{h}_\star^\top \mathbf{A}_\mathcal{M}^{-1} \mathbf{h}_\star + \sigma^2). \quad (25)
$$

which directly leads to the expression of the information criterion

$$
\begin{aligned}
\mathrm{MIC}_\eta((\mathbf{x}_\star, y_\star); \mathcal{M}) =& \eta \log \mathcal{N}(y_\star|\mathbf{h}_\star^\top \mathbf{A}_{\mathcal{M}^+}^{-1} \mathbf{b}_{\mathcal{M}^+}, \sigma^2 \mathbf{h}_\star^\top \mathbf{A}_{\mathcal{M}^+}^{-1} \mathbf{h}_\star + \sigma^2) \\
&- \log \mathcal{N}(y_\star|\mathbf{h}_\star^\top \mathbf{A}_\mathcal{M}^{-1} \mathbf{b}_\mathcal{M}, \sigma^2 \mathbf{h}_\star^\top \mathbf{A}_\mathcal{M}^{-1} \mathbf{h}_\star + \sigma^2).
\end{aligned}
$$

We can now combine this with the definition of the log density

$$
\log \mathcal{N}(y|a, b) = \log\left(\frac{1}{b\sqrt{2\pi}} e^{-\frac{1}{2}\left(\frac{y-a}{b}\right)^2}\right) = -\frac{1}{2}\left(\left(\frac{y-a}{b}\right)^2 + \log(2\pi b^2)\right)
$$

of a Gaussian $\mathcal{N}(a, b)$. Let us denote with $\mu_+$ and $\sigma_+$ the updated mean and variance, and with $\mu_-$ and $\sigma_-$ the old mean and variance (not to be confused with the noise variance $\sigma$. Then

$$
\mathrm{MIC}_\eta((\mathbf{x}_\star, y_\star); \mathcal{M}) = -\frac{\eta}{2}\left(\left(\frac{y_\star - \mu_+}{\sigma_+}\right)^2 + \log(2\pi\sigma_+^2)\right) + \frac{1}{2}\left(\left(\frac{y_\star - \mu_-}{\sigma_-}\right)^2 + \log(2\pi\sigma_-^2)\right)
$$

$\square$

