# OpenReview forum: "Information-theoretic Online Memory Selection for Continual Learning"
_ICLR.cc/2022/Conference — ICLR 2022 Poster_

### Official Review · Reviewer_6ZWf · 2021-10-18

**Correctness:** 3
**Technical Novelty And Significance:** 2
**Empirical Novelty And Significance:** 2
**Recommendation:** 6
**Confidence:** 4

**Main Review:**

Objective of the work:
* The paper extends the existing research direction in general continual learning. The key contribution is proposing the criteria to select and replace the existing points in the memory buffer.

Strong points:
* A new method for online memory selection. The surprise captures how unexpected a new point is given the memory, and allows us to include new information in the memory. The learnability captures how much of this new information can be absorbed without interference, allowing us to avoid outliers.
* The proposed Memorable Information Criterion (MIC) is a weighted combination of the proposed learnability and surprise scores. The authors also connect that MIC is related to Information Gain. In the appendix, the paper has compared these criteria (MIC, IG, ER) thoroughly.

Weak points:
* The surprise score is related to the uncertainty score in literature which has been widely used to select unseen points. While the learnability makes sense to distinguish the unfamiliar and outlier points. Having said that, the reviewer thinks that the learnability score can make the algorithm too preservative to learn new information.

* The performance may critically depend on balancing how much we want to “surprise” versus “learn” through $\eta$. It is highly recommended to have the ablation study with respect to $\eta$.

* Bayesian linear regression requires matrix inversion of a matrix $d \times d$ where $d$ is the feature dimension. The Sherman-Morrison Formula, Eq. (11), can be used to efficiently update the inverse matrix. However, while the reviewer acknowledges the trick works well in practice, the reviewer doesn't see this is a great technical contribution of the paper.


Presentation:
* The paper is in general well written and easy to follow.

Reproducibility:
* The results appear to be reproducible though I would encourage the authors to release their code to enable better validation of their claims.


Minor points:
* The author has demonstrated the effectiveness of the proposed criteria using a greedy algorithm to replace the existing points with the new points if certain condition is met. It could be interesting to consider either replacing (as the proposed framework) versus expanding without replacing. The reviewer thinks that we may need to enlarge the memory buffer overtime to learn more tasks. However, this would be out of the scope of the paper.

* In Figure 2, it is not clear to compare the middle figure and right figure to see the effect of learnability.

* The paper presents a scalable Bayesian model that can compute surprise and learnability with a small computational footprint by exploiting rank-one matrix structures. Instead of using a deep network and Bayesian linear model, I am wondering if it is better to directly train a Bayesian neural network for estimating the uncertainty?

* a footnote must be placed at the end of a clause or sentence, such as adding the number after the comma or full stop.



**Summary Of The Paper:**

This paper considers an online memory selection task for continual learning applications. The paper proposes a new (information-theoretic) criterion to pick informative points and void outliers. This criterion is a combination of the surprise and learnability score. In addition, the paper presents a stochastic information-theoretic reservoir sampler (InfoRS) to sample among selective points with high information.

The paper empirically demonstrates the proposed criteria encourage to select representative memories for learning the underlying function. They consider continual learning benchmark as well as ablation studies of data imbalance.


**Summary Of The Review:**

The key contribution is proposing the criteria to select and replace the existing points in the memory buffer. However, the technical novelty is limited. The paper includes several techniques which have been widely used elsewhere. The surprise (uncertainty estimation) has been widely used in literature. The Bayesian linear regression and the rank-one update trick are also well studied.

Empirically, the performance may critically depend on balancing how much we want to “surprise” vs “learn”. This should be well studied in the experiments. Perhaps, the paper can propose a technique to adaptively learn $\eta$ over time.

The reviewer thinks the paper is currently at the borderline.

---

> ### Author Response · Authors · 2021-11-15
> **Author Responses**
>
> Thank you very much for your detailed reviews and constructive comments. We have included new experiments trying to address your questions. Here are our responses.
>
> **Novelty.** Firstly, the authors think both the information-theoretic criteria and the efficient Bayesian model are important contributions for online memory selection in continual learning. The surprise criterion has been used before, highlighting our contribution in going beyond it with learnability to avoid outliers. The Bayesian model uses existing techniques, but combining them in online memory selection is non-trivial. Furthermore, the InfoRS algorithm forms another important contribution, which presents a simple way to adopt the information-theoretic criteria and avoid the identified timing issue. Finally, we note that the algorithm makes a significant practical advance in continual learning. Delivering a diverse memory using a fast algorithm is important in online memory selection. Previous methods such as GSS [1] and Coreset [2] incur prohibitively large computational overheads, inapplicable to even modest size problems. Reservoir sampling runs efficiently but suffers from data imbalance. As far as we know, our algorithm is the first approach that improves over RS while incurring negligible computational overheads even in the ResNet scale. Overall, we think that our paper makes solid contributions to the community.
>
> **Detailed Responses**
> 1. **Learnability and the Ablation study of $\eta$.** We note that incorporating learnability is critical to avoid outliers in online memory selection. Firstly, the TSNE plot in Figure 2 highlights the importance of learnability in avoiding outliers (A detailed explanation of Figure 2 is presented below). Furthermore, we present an ablation study of the learnability ratio $\eta$ as shown in A of General Responses. Setting $\eta=0$ removes the learnability in InfoRS, which we observe leads to degraded performances. We also note that the learnability will make the algorithm too preservative to learn new information only if $\eta$ is very large, as shown in the ablation. Therefore, it is essential to incorporate learnability appropriately in online memory selection. Finally, we think the reviewer’s idea of adapting $\eta$ online can be an interesting future work.
> 2. **Contribution of the efficient Bayesian model.** We agree that both the Bayesian model and the Sherman-Morrison formula existed before in the literature. However, we argue that their combination is an important contribution to online memory selection in continual learning. Particularly in online memory selection, where the memory buffer updates by one example at a time, their combination can be helpful to deliver running efficiency. We don’t think it is a trivial contribution.
> 3. **Enlarging the buffer along with learning.** Thank you for the suggestion. Enlarging the buffer along with the learning process is an interesting idea. In real problems, the memory constraint might not be strict, so it is acceptable to gradually enlarge the buffer, which won’t suffer from the issue of deleting previous memories. Our proposed information-theoretic criteria can help this process as well. We leave this as a future direction.
> 4. **Explain Figure 2.**  In Figure 2, we visualize the training examples (coloured dots) and the selected memories (black digits) using TSNE plots. For Figure 2(middle), which does not use learnability, we observe that many selected memories differ with training points in the same region. For example, in the orange area, which belongs to training points of class 4, memories from class 0,1,4,5,8 are selected. Thus the algorithm selects many outliers. In comparison, for Figure 2(right), which uses learnability, we observe that the selected memories usually accord with training points. For example, the memories in the orange area are all from class 4. This comparison demonstrates that learnability avoids outliers and encourages selecting representative memories.
> 5. **Using a Bayesian neural network.** Thanks for the suggestion. The proposed criteria are general enough to be combined with any Bayesian model, including a Bayesian neural network. Nevertheless, evaluating surprise and learnability will be intractable when using a Bayesian neural network. It might be possible to estimate them using techniques such as annealed importance sampling [3] or other log-partition estimators. However, running the estimators without sacrificing computational efficiency needs further investigation.
> 6. **Footnote.** Thanks for the suggestion. We have corrected that.
>
> [1] Aljundi et. al., Gradient-based sample selection for online continual learning
>
> [2] Borsos et. al., Coresets via Bilevel Optimization for Continual Learning and Streaming
>
> [3] Neal, annealed importance sampling.

---

> > ### Comment · Reviewer_6ZWf · 2021-11-22
> > **Response to authors**
> >
> > Thanks for the responses. I have read the response and will keep my score of 6.

---

### Official Review · Reviewer_TW8M · 2021-10-28

**Correctness:** 4
**Technical Novelty And Significance:** 3
**Empirical Novelty And Significance:** 3
**Recommendation:** 5
**Confidence:** 3

**Main Review:**

Strengths:

+ The proposed methods seem fairly natural. In particular, the notion of surprise and learnability seem well-suited to the current problem at hand. The computation of the Bayesian posteriors is also natural and the Gaussian assumption, together with conjugacy, results in efficient updates, which is nice.

+ The InfoRS algorithm also seems natural for sampling.

Weaknesses:

- It would be preferable for the authors to consider the effects of each of their proposed modules separately -- surprise, learnability as well as InfoRS. Currently from the experimental section, it is unclear which module contributes to which aspect of the improvement. An ablation study would be useful.

- I am wondering whether the authors have examined the effect of the many hyperparameters in their study such as \eta, \alpha and \beta.

- The datasets used seem rather simplistic, just MNIST and CIFAR10.

**Summary Of The Paper:**

This paper considers task-free continual learning in the context of online memory selection. The author consider the use of information-theoretic principles and propose the surprise and learnability criteria to select informative points and avoid outliers. The authors also propose InfoRS to sample among selective points with high information. The methods are validated on continual learning benchmark datasets, and show efficiency and improvements over existing approaches.

**Summary Of The Review:**

Generally, the paper has some novel contributions but also some minor weaknesses. If the authors can address the weaknesses/queries I raised in the main review, I'm happy to increase my score.

---

> ### Author Response · Authors · 2021-11-15
> **Author Responses**
>
> Thank you very much for your constructive comments. We have run additional experiments trying to address your questions. Here are our responses.
>
> 1. **Ablation study of surprise, learnability, and $\eta$.** Thank you for your suggestion. The learnability ratio $\eta$ controls the weighting between surprise and learnability. Specifically, setting $\eta=0$ removes learnability in InfoRS.  As mentioned in A of General Responses, we added an ablation study of $\eta$. The ablation demonstrates that balancing learnability and surprise is important in the selection.
> 2. **Ablation study of InfoRS.** Thanks for the suggestion. We conducted an ablation study of InfoRS in Figure 9 in the appendix. Specifically, we compared InfoRS (stochastic add, stochastic removal), InfoGS (deterministic add, deterministic removal), and another proposed InfoGS-RS (stochastic add, deterministic removal). We observe that InfoRS outperforms both InfoGS and InfoGS-RS. The ablation demonstrates that exploiting the information-theoretic criteria in an appropriate algorithm is important to online memory selection, which shows our contribution in proposing the InfoRS algorithm.
> 3. **Effect of $\alpha$ and $\beta$.** We note that $\alpha$ and $\beta$ are hyperparameters of DER++ [1] regarding how to update the predictor. However, our paper focuses on online memory selection, and we can combine the proposed approaches with any predictor updating algorithms. Therefore, studying the effect of $\alpha$ and $\beta$ is not within our scope.
> [1] Buzzega et. al., Dark Experience for General Continual Learning: a Strong, Simple Baseline
> 4. **Dataset.** Please note that we used standard continual learning benchmarks in experiments. And our experiments include Split MiniImageNet as well, beyond MNIST and CIFAR10. We believe the empirical results show strong evidence about the effectiveness of the proposed approach.

---

> > ### Comment · Reviewer_TW8M · 2021-11-18
> > **Response to authors**
> >
> > Dear authors,
> > Thanks for your response to my questions. Indeed, the ablation studies seem to be necessary for this line of works, and should be included if the paper is accepted (or included in a new version should the paper be rejected). However, even if $\alpha$ and $\beta$ are not the crux of your work, can you comment on how sensitive the performance is to the choices of $\alpha$ and $\beta$?
> > Thanks,
> > Reviewer

---

> > > ### Author Response · Authors · 2021-11-20
> > > **Author Response**
> > >
> > > Thank you very much for the additional response. To answer your question on the sensitivity of the proposed approach to $\alpha$ and $\beta$, we add another experiment comparing the performance of InfoRS under various $\alpha$ and $\beta$. The results are shown in Figure 13 in the updated paper. The new experiment shows that the proposed approach is not sensitive to the choice of $\alpha$ and $\beta$.

---

### Official Review · Reviewer_PFQ2 · 2021-11-02

**Correctness:** 4
**Technical Novelty And Significance:** 3
**Empirical Novelty And Significance:** 3
**Recommendation:** 8
**Confidence:** 4

**Main Review:**

Strengths:

1. I think that the paper was easy to follow overall, and the MIC criterion is intuitive. The tSNE plots (of the selected example for the memory of size 200) showing the ablation study was also quite interesting.
2. The proposed algorithm outperformed RS and other mentioned baselines, without too much computational overhead over RS.
3. Authors do study over other possible metrics than MIC, including Entropy Reduction, therefore empirically justifying the criterion.

Weakness: This subsection is a collection of doubts, suggestions about some more experiments (including other baselines), and how this submission can be improved overall.

Baselines and Performance metrics:
1.  I suggest including the paper [1] and online k-medoid clustering. [1] “Coresets via Bilevel Optimization for Continual Learning and Streaming” NeurIPS’20.
2. In my opinion, it is also important to report the backward transfer or forgetting metric, similar to the DER paper.
3. While the manuscript contains final testing results for different imbalance levels, I think for CL papers a plot of accuracy after each task provides more information ( For example, see Fig 3 in the A-GEM paper -  EFFICIENT LIFELONG LEARNING WITH A-GEM).

Technical Questions:

1. How sensitive is the performance with the thresholding hyperparameter?
2. Fig (5) right, similar to infoRS, I think for RS one also needs the confidence bars.
3. In the section about robustness to imbalance, does a different seed change the task ID in which we add more epochs? Moreover (more of a sanity check), but do authors keep the task-iD same across different “R” values?
4. Algorithm (2): What is the need to compute A_M^-1 and b_M when the memory is not full? Cannot one just do that once we fill in the memory, and use features based on the new parameters?
5. Algorithm (2): Rather a technical question - why are we only choosing one example from each batch? There can be a case that we have two examples with similar (but not the same) and high MIC values and are learnable (two different classes, right when we begin a new task).
6. Regarding GSS: Are authors using the GSS-IQP, or the GSS-Greedy? Moreover, one easy trick one can do to accelerate GSS is to consider gradient with respect to the last layer only, the way it is done in the paper [1] (See [2] as well). [1] Coresets for Data-efficient Training of Machine Learning Models, ICML’20. [2] Not all samples are created equal: Deep learning with importance sampling, ICML’18.
Regarding GSS: Rather a sanity check, but do authors use DER++ loss in this case for the fair comparison?
7. In Section E appendix (Proof), equation (19), where the KL is expanded, in my opinion, the numerator seems notationally odd. That is, why is x_{\ast} mentioned as a conditioned variable and as a parameter at the same time. Wouldn’t it suffice to just use it as a conditioned variable?

Technical suggestions:
1. Fig (2), please show how does the memory look like in the tSNE plot for the cases with small sizes such as 10/25 for RS and infoRS (w/ and w/o learnability)
2. What is mentioned here as DER is actually DER++ in the original paper.
3. In the related works section, which talks about dataset distillation, there is another paper [1] (ICLR’21) that also talks about dataset distillation with applications in continual learning. [1] Dataset Condensation with Gradient Matching.

Writing Suggestions:

1. In equation (1), I think it will be good to not use \theta for both h and g.
2. Equation (3), during the overall update step, we update memory as well as the network parameters. Therefore, I suggest using notations accordingly.
3. In the algorithm block, I would recommend adding line numbers, which improve readability and cross-referencing in a discussion.
4. Section 2.1, last paragraph: Add spacing after the period right before the “Furthermore”.


**Summary Of The Paper:**

This work targets the problem of Continual Learning (CL) and In particular, the problem of populating the buffer memory in the Experience Replay (ER)-based paradigm. In ER, we are allowed to store a small subset of the incoming data stream, therefore deciding on a good policy to populate the memory buffer is important. The paper proposes MEMORABLE INFORMATION CRITERION (MIC) which is a combination of the surprisal and learnability of an incoming example and further motivates this with a  bayesian analysis. Using MIC, it introduces two algorithms, namely, InfoRS and infoGS for buffer update. The paper includes experiments on standard CL datasets with a focus on General Continual Learning (GCL) guideline, which focuses on not having dependence on task boundaries. Finally, it reports the proposed method's performance on imbalanced data streams and shows that infoRS outperformed reservoir sampling, GSS, and some other baselines.


**Summary Of The Review:**

It is an interesting work, however, lacks discussion about some baselines (not mentioned in the paper) and other performance metrics (about forgetting, BWT). There are some questions (mentioned in the detailed review) that need to be answered in order to make it a strong submission.

---

> ### Author Response · Authors · 2021-11-15
> **Author Responses**
>
> We appreciate your valuable and constructive comments. We have included new experiments following your suggestions. Below are our responses,
>
> ### Baselines and Performance metrics:
> We have included them in the new revision, as mentioned in B, C, D of General Responses.
>
> ### Technical Questions:
> 1. **Sensitivity of the thresholding hyperparameter.** The thresholding parameter $\gamma_i$ determines the degree of InfoRS from acting purely randomly to greedily. We present an ablation as discussed in A of General Responses.
> 2. **Confidence bar of RS.**  We double-checked it. The figure has confidence bars, but the bar of RS is hidden due to the line width  (i.e. the error bar is smaller than the line width).
> 3. **Choosing Task-ID.** We deliberately tie the task-ID that receives more epochs with the random seed so that the imbalance appears for all tasks across the random runs. For example, in Split-CIFAR10, we make sure each of the five tasks is chosen twice within ten random runs. We have made this clearer in the experimental section in the appendix.
> 4. **The InfoGS Algorithm.** Thanks for the catch. It is unnecessary to compute $A_m^{-1}$ and $b_m$ before the memory is full.
> 5. **The InfoGS Algorithm.** InfoGS is not limited to choosing one example only within a batch, as seen in the FOR loop in the algorithm. Given the whole batch, InfoGS loops over selections. In each step, InfoGS attempts to replace the least informative memory point with the most informative batch point. InfoGS jumps out of the loop only when the replacement fails or the replacement happens for $|\mathcal{B}|$ times. Empirically we observe that InfoGS tends to jump out of the loop quickly, which results in its efficiency.
> 6. **GSS.** We used the GSS-Greedy. Thanks for the suggestion, considering only the last layer weights can be helpful for acceleration. Finally, we note, we used the DER++ loss for all methods to train the network for a fair comparison.
> 7. **The Proof Typo.** Thanks for the catch. It is a typo.
>
> ### Technical suggestions:
> 1. **The TSNE plot for the memory with a smaller buffer.** We have added it. Please refer to E of General Responses.
> 2. **DER++.** Yes, the loss is DER++ indeed. We have corrected it.
> 3. **Reference.** Thank you for pointing to this paper. We note that dataset distillation is tightly related to memory selection. We have added a reference for it. [1] Dataset Condensation with Gradient Matching.
>
> ### Writing Suggestions:
> Thanks for your suggestions. We have incorporated them into the paper.

---

> > ### Comment · Reviewer_PFQ2 · 2021-11-23
> > **Can authors cite and discuss the following paper?**
> >
> > The generalized algorithm for infoRS is somewhat similar to Algorithm 1 in [1]. This paper uses a fixed window instead of moving average and uses submodular function gain, instead of MIC.
> >
> >
> > [1]A Practical Online Framework for Extracting Running Video Summaries under
> > a Fixed Memory Budget (Lavania, Wei, Iyer and Bilmes ) https://epubs.siam.org/doi/pdf/10.1137/1.9781611976700.26

---

> > > ### Author Response · Authors · 2021-11-23
> > > **Author Response**
> > >
> > > Thank you for bringing this to our attention. It is closely related to the online memory selection problem we are looking at.
> > >
> > > This paper [1] investigates online submodular maximization [2] in the application of online collecting video summaries. Specifically, online submodular maximization [2] aims to select a memory buffer online that maximizes a submodular criterion on the buffer.  [2] present a greedy algorithm that includes a point if adding it improves the submodular criterion by at least a certain threshold, which is similar to the information improvement thresholding procedure in our proposed InfoGS. [1] extends the greedy algorithm and designs a submodular function to evaluate the collected video summaries.
> > >
> > > Given the similarities, we also point out two differences.
> > >
> > > 1. We focus on rehearsal-based continual learning, where online memory is also used to regularize network learning. In this case, a greedy algorithm can suffer from the issue of timing to update the memory as we pointed out in Sec 4. This motivates us to introduce the stochastic InfoRS.
> > >
> > > 2. While we provide MIC as a general criterion to evaluate forthcoming data points, [1, 2] do not provide a general approach to find meaningful submodular functions to evaluate the buffer. We note that the total entropy we compared in Sec C might be a promising choice.
> > >
> > > Finally, we have included this discussion in the revised paper.
> > >
> > > [1] Lavania et al, A Practical Online Framework for Extracting Running Video Summaries under a Fixed Memory Budget
> > >
> > > [2] Buchbinder et al., Online submodular maximization with preemption.

---

### Author Response · Authors · 2021-11-15
**General Responses**

Thank all the reviewers for the detailed responses and constructive suggestions. We have included new experimental results, corrected typos, and clarified experimental details. These changes are already incorporated into our paper. In particular, the new experimental results are listed in the following,

**A, Ablation study of $\eta$ and $\gamma_i$.** We report an ablation study for the learnability ratio $\eta$ and the information thresholding ratio $\gamma_i$ in Figure 10. Firstly, for the learnability ratio $\eta$, we observe setting $\eta$ to be small and setting $\eta$ to be large both lead to degraded performance. This ablation demonstrates the importance of balancing learnability and surprise in online memory selection. Secondly, $\gamma_i$ determines the degree of InfoRS from acting purely randomly to greedily. Its ablation demonstrates how InfoRS properly balances the algorithm’s stochasticity and greediness to counter different data imbalances. Finally, we observe that, for all data imbalance (R=1,3,10,30), the optimal $\eta$ are similar and the optimal $\gamma_i$ are similar, which indicates that InfoRS is not sensitive to these hyperparameters.

**B, Forward and backward transfer.** We have reported the forward transfer and backward transfer for all approaches in Table 3 and Table 4, respectively.

**C, Online Clustering and Coreset.** For an additional baseline, we include the feature-space clustering (FSS-Clust in [1]) for Permuted MNIST and Split MNIST. The test performances are included in Table 2. We observe that FSS-Clust underperforms RS and InfoRS. In addition, we modified the code of the streaming coreset selection (Coreset) [2] to match our experimental setting. However, because the Coreset method needs to solve multiple (12 in [2]) bilevel optimization problems in each iteration, it incurs a very large computational overhead (e.g., for Split MNIST without data imbalance, the Coreset method takes ~3 hours whereas RS and FSS-Clust take ~2 minutes). Therefore, although we tried very hard to improve its running efficiency, we haven’t finished the hyper-parameter search and haven’t got a result yet, not even for the smallest Split MNIST.

[1] Aljundi et. al., Gradient-based sample selection for online continual learning

[2] Borsos et. al., Coresets via Bilevel Optimization for Continual Learning and Streaming

**D, Evolution of test accuracies along with training.** We plot the evolution of average test accuracies along training in Figure 11.

**E, TSNE of a small memory.** Following the experiment setup in Figure 12, we add a TSNE plot with a small memory. Specifically, we set M=25 for the 10-way classification. We observe that InfoGS without learnability performs similarly to InfoGS with learnability. Intuitively, the surprise will dominate when the memory buffer is small. Then incorporating learnability is not critical to the selection.

---

> ### Comment · Reviewer_PFQ2 · 2021-11-16
> **Thanks for the tSNE plots and additional baselines.**
>
> Given the time constraint, I am fine with not having Coreset based baseline at this point, however, if the paper gets accepted, then I would like the paper to include  [2].
>
> It is a bit strange to me that GSS significantly underperforms FSS for simple split MNIST case, which is opposite to what [1] had in Table 1. Perhaps, a possible explanation could be that [1] used a buffer size of 300, v/s this paper used 100, and hence GSS might require a higher buffer size.
>
> Thanks for the tSNE plots.

---

> > ### Author Response · Authors · 2021-11-16
> > **Author Responses**
> >
> > We really appreciate your quick response.
> >
> > In terms of GSS, we comment on additional experimental differences between our paper and the GSS paper [1],
> > 1. [1] uses a subset of MNIST including 500 examples for each class, whereas we use the whole MNIST dataset.
> > 2. [1] uses experience replay (ER) for regularization (i.e., regularizing only the labels with $\alpha=0$), whereas we use DER++ and tune both $\alpha$ and $\beta$.
> >
> > We think these factors might as well contribute to the difference. In particular, DER++ includes the MSE regularization on logits compared to ER. Because FSS-Clust keeps diverse logits via online clustering in the logit space, it might provide better regularization signals in DER++ compared to GSS.

---

### Decision · Program_Chairs · 2022-01-20

**Decision:**

Accept (Poster)

**Comment:**

One way of avoiding catastrophic forgetting in continual learning is through keeping a memory buffer for experience replay.  This paper addresses the problem of online selection of representative samples for populating such memory buffer for experience replay.  The paper proposes novel information-theoretic criteria that selects samples that captures surprise (samples that are most informative) and learnability (to avoid outliers).  They utilize a Bayesian formulation to quantify informativeness. They provide two algorithms: a greedy approach, and an approach that takes timing (when to) update memory into account based on reservoir sampling to mitigate possible issues with class imbalance.

Pros:  The paper is well written and organized.  It was easy to follow. The formulation is novel and technically sound. The idea of taking learnability into account is novel and interesting.  It provides a nice way of avoiding outliers and balancing surprising information. The authors presented the motivation for each part of the framework well.

Cons: To understand the contribution of each component of the formulation and competing criteria, an ablation study is needed.
Reviewers had several detailed suggestions and questions, including sensitivity to hyperparameters, additional citations, additional data sets beyond MNIST and CIFAR10, etc.

In the rebuttal, the authors have addressed several of these concerns.  Please make sure to include and incorporate reviewer suggestions in the final revised version.